# Single allele loss-of-function mutations select and sculpt conditional cooperative networks in breast cancer

Nathan F. Schachter [1,2], Jessica R. Adams [1,2], Patryk Skowron[3,4,5], Katelyn. J. Kozma[1,2], Christian A. Lee[6,7], Nandini Raghuram[1,2], Joanna Yang [1,2,17], Amanda J. Loch[1], Wei Wang [1], Aaron Kucharczuk [1,2], Katherine L. Wright[1,2], Rita M. Quintana[1,18], Yeji An [1,2], Daniel Dotzko[1], Jennifer L. Gorman [8], Daria Wojtal [2], Juhi S. Shah[1], Paul Leon-Gomez [1], Giovanna Pellecchia[9], Adam J. Dupuy [10], Charles M. Perou [11], Ittai Ben-Porath[12], Rotem Karni [13], Eldad Zacksenhaus[5,14], Jim R. Woodgett[7,8], Susan J. Done [5,7,15,16], Livia Garzia[3,4,19], A. Sorana Morrissy [3,4,20], Jüri Reimand [2,6,7], Michael D. Taylor [3,4,5] & Sean E. Egan [1,2✉]

The most common events in breast cancer (BC) involve chromosome arm losses and gains. Here we describe identification of 1089 gene-centric common insertion sites (gCIS) from transposon-based screens in 8 mouse models of BC. Some gCIS are driver-specific, others driver non-specific, and still others associated with tumor histology. Processes affected by driver-specific and histology-specific mutations include well-known cancer pathways. Driver non-specific gCIS target the Mediator complex, $Ca^{++}$ signaling, Cyclin D turnover, RNA-metabolism among other processes. Most gCIS show single allele disruption and many map to genomic regions showing high-frequency hemizygous loss in human BC. Two gCIS, *Nf1* and *Trps1*, show synthetic haploinsufficient tumor suppressor activity. Many gCIS act on the same pathway responsible for tumor initiation, thereby selecting and sculpting just enough and just right signaling. These data highlight ~1000 genes with predicted conditional haploinsufficient tumor suppressor function and the potential to promote chromosome arm loss in BC.

[1] Program in Cell Biology, The Peter Gilgan Center for Research and Learning, The Hospital for Sick Children, Toronto, ON, Canada. [2] Department of Molecular Genetics, University of Toronto, Toronto, ON, Canada. [3] Program in Developmental & Stem Cell Biology, The Hospital for Sick Children, Toronto, ON, Canada. [4] The Arthur and Sonia Labatt Brain Tumour Research Centre, The Hospital for Sick Children, Toronto, ON, Canada. [5] Department of Laboratory Medicine and Pathobiology, University of Toronto, Toronto, ON, Canada. [6] Computational Biology Program, Ontario Institute for Cancer Research, Toronto, ON, Canada. [7] Department of Medical Biophysics, University of Toronto, Toronto, ON, Canada. [8] Lunenfeld-Tanenbaum Research Institute, Sinai Health System, Toronto, ON, Canada. [9] The Center for Applied Genomics, The Hospital for Sick Children, Toronto, ON, Canada. [10] Department of Pathology, Carver College of Medicine, The University of Iowa, Iowa City, IA, USA. [11] Lineberger Comprehensive Cancer Center, Departments of Genetics and Pathology, University of North Carolina, Chapel Hill, NC 27599, USA. [12] Department of Developmental Biology and Cancer Research, Institute for Medical Research-Israel-Canada, The Hebrew University-Hadassah Medical School, Jerusalem, Israel. [13] Department of Biochemistry and Molecular Biology, Institute for Medical Research Israel Canada (IMRIC), Hebrew University-Hadassah Medical School, Jerusalem, Israel. [14] Division of Cell and Molecular Biology, Toronto General Research Institute, University Health Network, and Department of Medicine, University of Toronto, Toronto, ON, Canada. [15] The Princess Margaret Cancer Centre, University Health Network, Toronto, ON, Canada. [16] The Laboratory Medicine Program, University Health Network, Toronto, ON, Canada. [17]Present address: Faculty of Medicine, University of Toronto, Toronto, ON, Canada. [18]Present address: Natera, San Francisco, CA, USA. [19]Present address: Cancer Research Program, McGill University, Montreal, QC, Canada. [20]Present address: Department of Biochemistry and Molecular Biology, University of Calgary and Arnie Charbonneau Cancer Institute, Calgary, AB, Canada. ✉email: segan@sickkids.ca

Analysis of thousands of human tumors has led to the identification of recurrently mutated oncogenes and tumor suppressors (TSG)[1]. In breast cancer, 99 such genes are thought to play a particularly important role[2–4]. Despite the presence of focal alterations in dominant oncogenes and recessive TSGs, more frequent changes to the breast cancer genome involve losses and gains of large regions, often at the level of entire chromosomal arms[5]. Many of the deletions are now thought to select for loss of haploinsufficient tumor suppressor genes (hTSGs) that promote tumor initiation and/or progression when hemizygous[6]. In this regard, a cancer gene island model has been proposed, whereby commonly deleted regions contain many genes that reduce proliferative fitness but few that promote proliferation[7]. Since growth suppression is but one of ten cancer hallmarks[8], it would seem likely that copy number alteration (CNA) impact multiple cancer cell properties in a context-dependent manner.

Early screens to identify genes with the potential to induce mammary tumors in mice involved mouse mammary tumor virus (MMTV)-induced insertional mutagenesis. Indeed, several large-scale MMTV screens have been performed, with most common insertions generating potent gain-of-function alleles[9,10]. More recently, Sleeping Beauty (SB) transposon screens have been described, which can both inactivate and activate target gene function. In contrast to MMTV-based screens, however, most gene-centric Common Insertion Sites (gCISs) in SB screens appear to represent loss-of-function alleles, affecting one copy of the target gene[11]. SB screens, therefore, provide a very effective approach to identify genes that promote tumor formation in a hemizygous state. In addition, these screens can involve mobilization of dozens of transposons within each cell. As a result, they can be used to identify cooperative interactions that promote transformation[12]. Several driver-specific SB screens have been performed in the mouse mammary gland, most based on activation of transposition in cytokeratin-5 expressing cells[11,13,14]. The Wap-Cre and MMTV-Cre systems have also been used to screen for SB insertions that promote mammary tumor formation[15,16]. To date, individual screens have been performed on sensitized backgrounds with activation of Ctnnb1 as well as deletion of Pten, Trp53, Cdh1, or Brca1[11,13–17].

In this work, we describe a large-scale systematic approach to transposon-based cancer gene discovery in mammary epithelium. We report results of Sleeping Beauty screens in mice from eight different models of breast cancer and in control FVB mice. The gCIS identified are predominantly driver-specific and most appear to represent single copy loss-of-function alleles. The gene sets identified herein show relatively little overlap with those showing focal mutation, amplification, or deletion in human BC and may therefore include many haploinsufficient tumor suppressors driving hemizygous loss of chromosome arm-length regions of the genome.

## Results

**Sleeping beauty mammary cancer gene discovery screens on multiple genetic backgrounds.** To identify genes involved in initiation or progression of BC, we performed SB transposon-based screens in mammary glands from eight different genetically engineered mouse model (GEMM) strains, each selected on the basis of activation of genes/pathways thought to promote BC ($Pik3ca^{E545K}$, $Pik3ca^{H1047R}$, $Trp53^{LSL-R270H}$, $Kras^{G12D}$, $Notch1^{ICD}$, Elf3, and Stat3C) or deletion of tumor suppressor genes which inhibit it (Lfng) (Supplementary Table 1). In each case, we induced Cre-dependent transposon mobilization together with oncogene activation or tumor suppressor gene deletion using MMTV-Cre$^{NLST}$ (Supplementary Fig. 1 and Supplementary Table 1), which is relatively inefficient but more mammary-specific than other MMTV-Cre transgenics[18]. Using

this line we were able to minimize the incidence of lymphoma in experimental animals, a common problem with MMTV based transgenic systems. In our screens, the $R26^{LSL-SB11}$ knock-in allele was used to direct expression of SB transposase in a Cre-conditional manner, and SB transposons were derived from T2Onc3a and T2Onc3b mice, which have SB concatemers on different donor chromosomes[19,20]. In parallel, initiating events were activated without SB to establish baseline rates of tumor penetrance and cancer growth kinetics on all eight genetic backgrounds. As expected, mammary tumors formed in most of these lines, even without SB mutagenesis. SB mutagenesis either reduced tumor latency and/or increased incidence in most GEMM tested (Supplementary Fig. 2). In control mice, and in most sensitized backgrounds, mammary tumor formation occurred much faster in T2Onc3a-cohort animals than in those with T2Onc3b (Supplementary Fig. 2). The reason for this is unclear. One potential explanation could be that the SB concatemer in T2Onc3a mice maps close to one or more cancer genes, in which case local hopping could enhance tumor formation in this line. Alternatively, if the T2Onc3b concatemer has been methylated or otherwise silenced, SB-mediated mobilization with resulting tumor formation, could be impaired.

Next, we performed gCIS analysis[21] to identify SB insertions driving mammary tumor growth in each cohort (Fig. 1, Supplementary Fig. 3 and Supplementary Data 1/2). We identified 50 clonal gCIS in control cohort mice, 32 in $Pik3ca^{E545K}$ cohort mice, 62 in $Pik3ca^{H1047R}$, 37 in $Trp53^{R270H}$, 60 in $K-Ras^{G12D}$, 42 in $Notch1^{ICD}$, 124 in Stat3C, 18 in Elf3 and 9 in $Lfng^{loxP/loxP}$ cohort tumors, respectively. Individual tumors had anywhere from zero to almost 50 identifiable gCIS. This latter number exceeds available SB transposons in T2Onc3a and T2Onc3b donor concatemers[19] and reflects identification of gCIS within multiple tumor subclones (see top x-axis in Fig. 1 and Supplementary Fig. 3). Some gCIS were present at clonal and subclonal read-levels in different tumors (see right hand y-axis in Fig. 1 and Supplementary Fig. 3).

**Most gCIS are driver-specific.** We next compared gCIS identified in each cohort. The most commonly targeted genes in $Pik3ca^{E545K}$ cohort tumors were Fbxw7, Lpp, and Zfp148, which code for proteins involved in ubiquitylation/destruction of oncoproteins (Fbxw7)[22], control of cellular invasion/migration (Lpp)[23] and control of cell cycle regulation/insulin secretion/Wnt signaling (Zfp148)[24], respectively. In $Pik3ca^{H1047R}$ tumors, Trps1 (which codes for a transcription factor involved in epithelial biology and lineage specification within the mammary gland), Kmt2c (or Mll3)(which codes for a chromatin regulator that methylates Histone H3 on lysine 4), and Nipbl (which codes for a protein that regulates chromatin organization/looping through cohesin loading) were most frequently targeted. KMT2C is the fifth most common focally mutated gene in human BC[4]. When KMT2C copy number loss is also considered, this gene is functionally hemizygous in 24% of breast cancers[25], and is mutated together with PIK3CA in a significant number of cases ($p < 0.001$)[25]. As most tumors with KMT2C mutations retain one wildtype copy of the gene, it seems likely that this gene is a hTSG in breast cancer as in acute myeloid leukemia[26]. Interestingly, the top clonal gCIS identified in $Pik3ca^{E545K}$ cohort tumors were distinct from those selected in $Pik3ca^{H1047R}$ tumors (Fig. 1). In fact, Fbxw7, Lpp, and Zfp148 did not appear on the list of clonal gCIS from $Pik3ca^{H1047R}$ tumors at all, nor were Trps1, Kmt2c, and Nipbl identified as clonal gCIS in $Pik3ca^{E545K}$ tumors (Supplementary Fig. 3). The most commonly targeted genes in $Trp53^{R270H}$ tumors were Met, Rasa1, and Trps1. These findings are consistent with published work on cooperation between Trp53 loss-of-function and activated

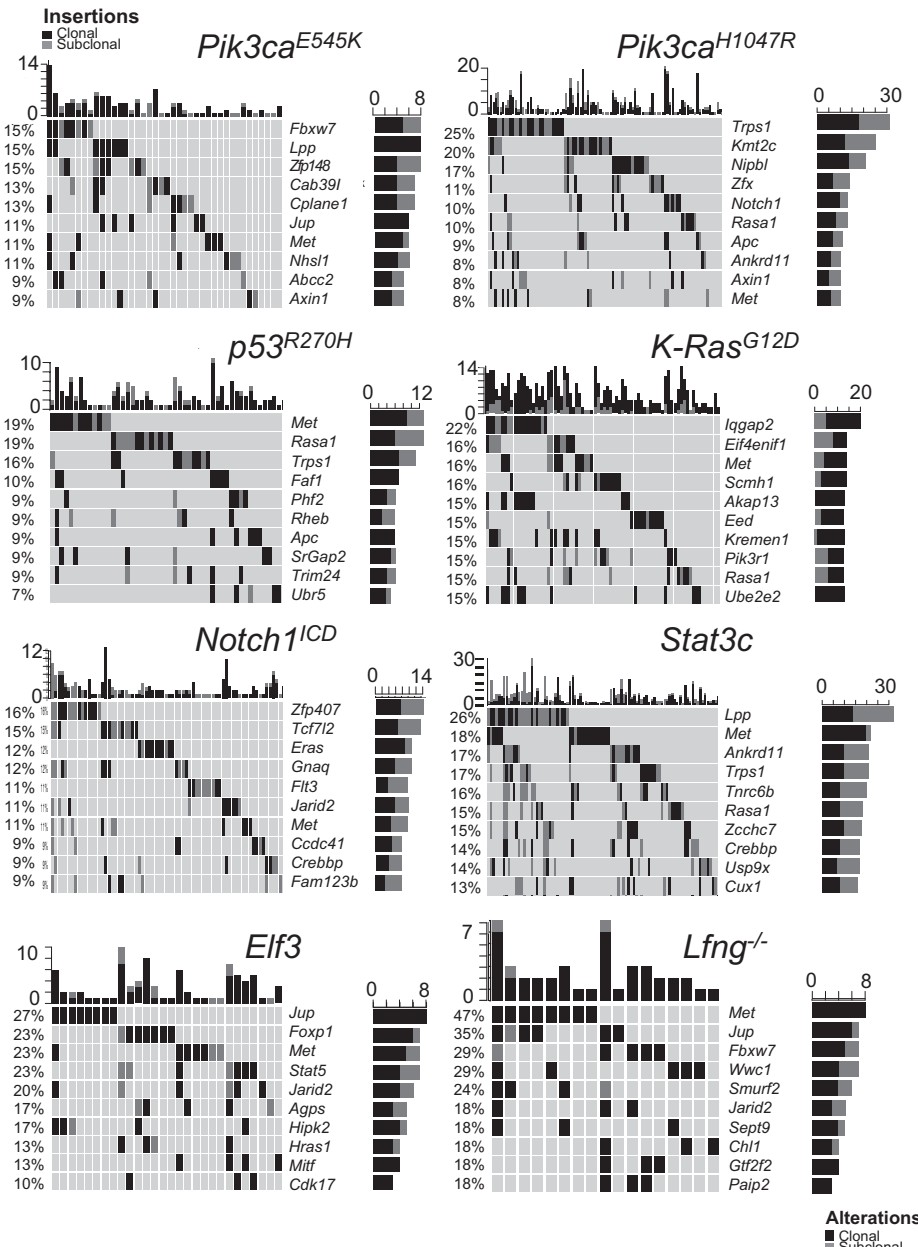

**Fig. 1 Top common insertion sites for Sleeping Beauty in tumors from distinct GEMM models of breast cancer.** Overview of clonal/subclonal gCIS from GEMM-based SB screens as indicated. Top 10 gCIS are shown for each. Note, the percentage of tumors with each gene targeted by SB is shown on the y-axis to the left for each bar graph, the number of tumors with clonal vs. subclonal SB targeting by SB is shown on the y-axis to the right, whereas the number of identified gCIS in each tumor is shown on the x-axis above each bar graph.

Met[27], as well as between hemizygous loss of Rasa1 and Trp53 mutation in mammary cells[13]. The most frequently targeted genes in each cohort were distinct for the most part, although Met, Trps1, Fbxw7, Jup, and Lpp appeared near the top of more than one cohort-specific list (Fig. 2a, Supplementary Fig. 3 and Supplementary Data 1/2).

The only clonal gCIS identified in all eight GEMM was Met, which codes for a receptor tyrosine kinase (Supplementary Data 1). In almost every case, Met appeared to be activated through increased transcription resulting from SB insertion at its 5′ end (Supplementary Fig. 4). Another common target was Jup, which encodes Plakoglobin. In this case, SB insertions clustered within intron 2 (Supplementary Fig. 4), and were predicted to generate a stabilized, oncogenic N-terminally truncated fragment of Plakoglobin. Jup was targeted and clonally selected in five

GEMM backgrounds (Pik3ca^{E545K}, Pik3ca^{H1047R}, Elf3, Stat3, and Lfng). In four backgrounds, Rasa1, Trps1, and Kat6a (Myst3) were targeted with bi-directional SB insertions that were dispersed across the entire length of each gene. This pattern of mutagenesis is consistent with loss-of-function. Seven genes were targeted in three backgrounds (Sp3, Fbxw7, Ankrd11, Nf1, Eif4enif1, Stat5a/b, Notch1) and 24 genes in two. In most cases, transposon insertions were found in both orientations and spread throughout the gene in question, likely indicative of gene disrupting insertions. Clear, but not exclusive, exceptions to this included Met and Jup, as noted above, as well as Stat5, Notch1, Flt3, and Eras (Supplementary Fig. 4). For Jup and Notch1, N-terminally deleted mutant fragments consistent with activation through SB-mediated truncation and overexpression are readily detected by western analysis (Supplementary Fig. 4).

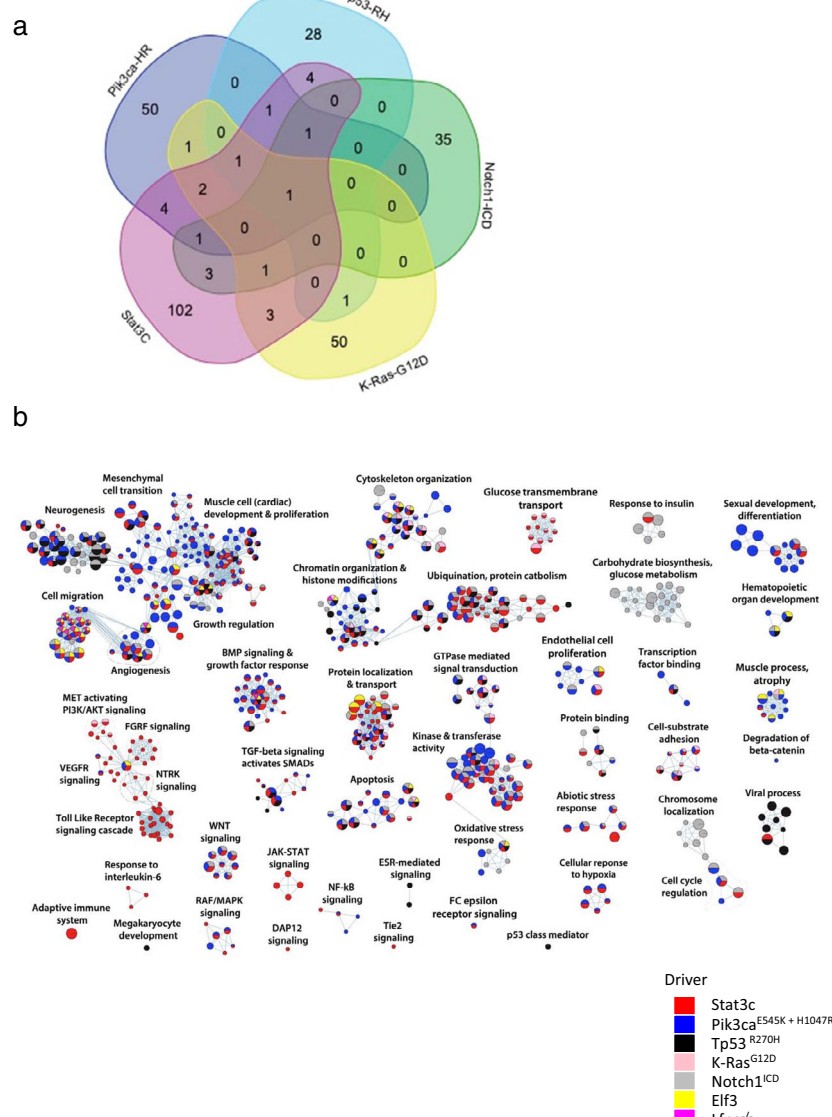

**Fig. 2 Pathway analysis identifies shared and driver-specific pathways and processes in transformation of mammary epithelium. a** Overlap of clonal gCIS identified in GEMM-specific cohorts. **b** Pathways altered by SB mutagenesis in GEMM-specific subscreens (FDR corrected *P*-value < 0.05). Enrichment map shows enriched pathways and processes as nodes, while pathways with many shared genes are connected and cluster into subnetworks. Node colors indicate cohorts where gCIS events were enriched in the corresponding pathways. One-sided ranked hypergeometric tests were used and significant pathways visualized. *p*-value information for nodes are found in Supplementary Data 4.

To identify gCIS that occur on multiple backgrounds (i.e., driver non-specific) but fall below the significance threshold in individual cohorts, we combined tumors from *Pik3ca^E545K*, *Pik3ca^H1047R*, *Trp53^R270H*, *K-Ras^G12D*, *Notch1^ICD*, *Stat3C*, *Elf3*, and control mice into a single large pan-mammary tumor cohort and repeated gCIS analysis. This combined cohort of ~800 tumors yielded 193 gCIS, 96 of which were not identified in driver-specific or control cohorts (Fig. 3a as well as Supplementary Data 1–3). Many of the 96 genes were targeted by SB in dozens of tumors (including clonal and subclonal insertions) but were not statistically significant when initiating event-specific cohorts were analyzed in isolation. For example, genes coding for Mediator complex subunits (*Med13* and *Med13l*) were frequently disrupted (64 and 33 tumors showed single allele disruption at either clonal or subclonal level, respectively). These genes, which map to chromosome 17q23.2 and 12q24.21 respectively, show hemizygous loss in ~11 and 13% of human BCs[25]. Indeed, genes coding for other Mediator subunits were identified in driver- and histology-specific cohorts (*Med26*

and *Med28*). *Ppp3ca*, which encodes a catalytic subunit of Calcineurin phosphatase, showed single allele disruption in a total of 48 tumors. Interestingly, this gene, which maps to chromosome 4q24 shows hemizygous loss in ~15% of human BCs[25]. The *Ambra1* gene, which coding for a Cyclin D targeting subunit of Cul4 containing CRL ligase was targeted in 37 tumors. This gene maps to 11p11.2 and shows hemizygous loss in ~13% of tumors[25]. These tumors show significant co-selection for RB1 hemizygous loss[28]. Genes coding for proteins involved in processes including but not limited to RNA metabolism (*Cnot6l*, *Pan3*, *Rbm9/Rbfox2*, and *Zfr*) were also identified (Supplementary Data 3). Some of these large cohort specific genes show hemizygous loss in over 40% of breast tumors.

**Initiating event-specific and common pathways/processes associated with mammary tumor formation.** Next, we performed pathway enrichment analysis on gCIS from each cohort[29].

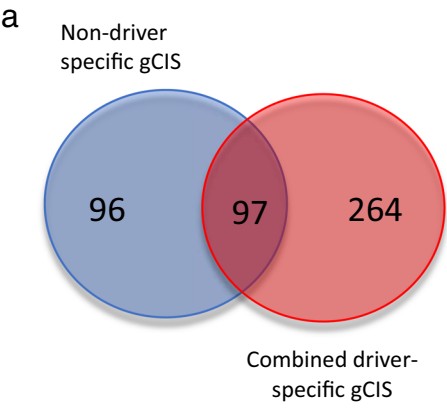

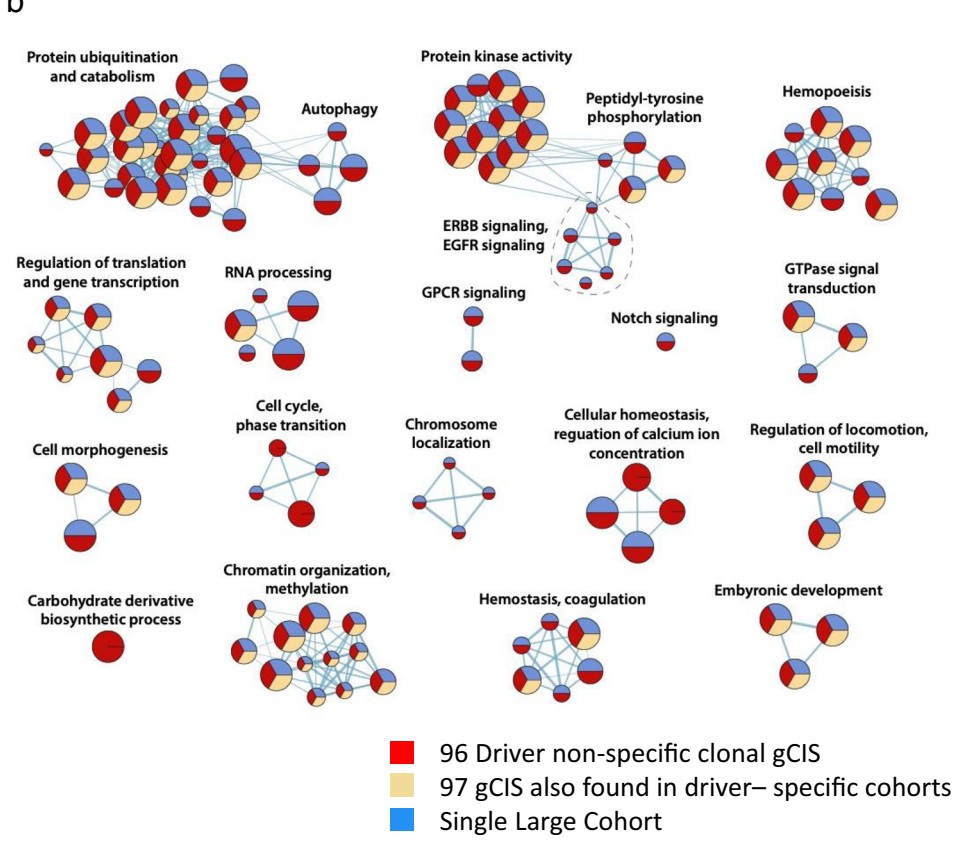

■ 96 Driver non-specific clonal gCIS
■ 97 gCIS also found in driver– specific cohorts
■ Single Large Cohort

**Fig. 3 SB insertion site analysis on one large cohort reveals distinct driver non-specific gCIS. a** Identification of 96 driver non-specific gCIS in large cohort of tumors from *Pik3ca^E545K*, *Pik3ca^H1047R*, *p53^R270H*, *Kras^G12D*, *Stat3C*, *Notch1^ICD*, and *Elf3* GEMM model mice, These gCIS were not identified when each model was analyzed in isolation. **b** Pathways altered by SB mutagenesis in driver non-specific large cohort screen—comparing gCIS that overlap with those found in driver-specific cohort with those only identified in the large combined cohort (FDR corrected *P*-value < 0.05). Enrichment map shows enriched pathways and processes as nodes, while pathways with many shared genes are connected and cluster into subnetworks. Node colors indicate cohorts where gCIS events were enriched in the corresponding pathways. One-sided ranked hypergeometric tests were used and significant pathways visualized. *p*-value information for nodes are found in Supplementary Data 4.

This revealed that a large fraction of gCIS coalesce around common signaling pathways and processes whose alteration were selected for by multiple initiating events (Fig. 2b). For example, genes coding for proteins connected to programmed cell death were enriched in six different GEMM cohorts, genes regulating chromatin organization in five, and genes facilitating or regulating RTK/MAPK and/or GTPase signal transduction in four (Supplementary Data 4). *Pik3ca* and *Stat3* tumors selected for mutations affecting TGFβ/BMP signaling, whereas the PI3K/AKT

signaling pathway was enriched in *K-Ras* and *Stat3C* tumors (Supplementary Data 4).

Interestingly, genes involved in canonical Wnt signaling were identified in three different cohorts (*Pik3ca^H1047R*, *Stat3C*, and *Notch1^ICD*), however, the genes identified in each were distinct (Fig. 2b as well as Supplementary Data 1 and 4). For example, in *Pik3ca^H1047R* tumors, SB targeted *Apc*, *Axin1*, *Ddit3*, and *Cul1* to create what appeared to be loss-of-function insertions, which should thereby activate Wnt signaling. In *Stat3C* tumors, SB

insertions were selected in *Tmem170b*, *Tnks*, *Znrf3*, *Smurf2*, *Kdm6a*, *Mapk14*, *Ppp2ca*, *Csnk2a1*, *Crebbp*, *Pten*, and *Cul1*. Some of these encode core inhibitors of the Wnt pathway (*Tmem170b*, *Tnks*, and *Znrf3*). Thus, the apparent loss-of-function mutations identified in each case should also activate signaling. Finally, in *Notch1*$^{ICD}$ cohort tumors, loss-of-function insertions were selected in *Tcf7l2*, *Gsk3b*, *Rnf43*, *Amer1* (also known as *Fam123b* and *Wtx*), *Zfp-148* (also known as *Zbp-89*), *Hdac6*, *Gnaq*, *Crebbp*, and *Pten*. In this case, predicted loss-of-function mutations in *Tcf7l2* and *Zbp-89*/*Zfp-148* (both of which encode positive elements in the Wnt signaling pathway) should reduce Wnt signaling, whereas predicted loss-of-function mutations in *Amer1* could enhance or suppress it[30]. These results suggest that distinct nodes within signaling pathways are targeted in each cohort, depending on which oncogenic initiating event has already been activated within the cell-of-origin. This phenomenon has been described before in a large-scale SB screen for gene insertions that promote gastrointestinal tract (GI) tumors across multiple GEMM[31].

Some pathways or processes were altered through SB-mediated mutagenesis in only one cohort (Fig. 2b and Supplementary Data 4). For example, groups of genes involved in carbohydrate metabolism (*Ppara*, *Gsk3b*, *Tcf7l2*, *Igf2*, and *Pten*) and chromosome localization (*Vps4b*, *Cenpq*, and *Cenpc1*) were only selected for in *Notch1*$^{ICD}$ tumors. The former group likely relate to a known connection between Notch and glucose metabolism in other tissues[32]. In contrast, groups of genes affecting FGFR signaling (*Ppp2ca*, *Nras*, *Kras*, *Gab1*, and *Pik3r1*) as well as Toll-like receptor and Interleukin signaling (*Mapkapk2*, *Ppp2ca*, *Cul1*, *Mapk14*, *Il6st*, and *Socs5*) were selected exclusively in *Stat3C* tumors. Next, we performed pathway analysis on the 96 driver non-specific gCIS identified exclusively when data were analyzed as part of one large cohort (as discussed above), revealing selection for insertions that affected autophagy, ErbB signaling, regulation of Ca$^{++}$ ion concentration as well as chromosome localization (Fig. 3b and Supplementary Data 4).

**gCIS identification on the basis of mammary tumor histology.** To determine whether mammary tumors of different histotype were linked to selection for distinct insertional events, we analyzed lesions using a classification system developed by Robert Cardiff and colleagues (Supplementary Fig. 5)[33]. Interestingly, SB-mediated mutagenesis changed the histological profile of tumors that formed in several GEMM (Supplementary Fig. 6 and Supplementary Data 5). To correlate tumor phenotype with disruption (or activation) of specific genes, tumors were grouped into five histological families: (i) Adenosquamous carcinomas, (ii) Papillary tumors, (iii) Poorly differentiated adenocarcinomas, (iv) Spindle tumors (a histological group including Spindle cell tumors and Scirrhous tumors), and (v) Squamous tumors (including Squamous cysts, Squamous cell carcinomas, and Keratoacanthomas). Insertion site data were then reanalyzed to identify gCIS associated with each mammary tumor histotype (Supplementary Data 1). 312 gCIS were identified in this way, 136 of which were not seen when tumors were analyzed as part of driver-specific cohorts or as one large cohort (see above)(Supplementary Data 3). Within these 136 gCIS, among other pathways and processes, we found genes that regulate RNA processing (*Cpsf7*, *Snd1*, *Srp72*, *Cpeb4*, *Xrn2*, and *Cnot1*) and small GTPase signaling (*Arl8b*, *Arf3*, *Arl9*, *Dennd5a*, *Tbc1d10a*, *Erc1*, and *Rab5b*).

As with driver-based analysis, some gCIS were identified in more than one histology-based cohort (e.g., *Trps1* single allele disruption was selected for in tumors from four histological families) (Fig. 4a/b, Supplementary Fig. 7 and Supplementary

Data 1). Despite the presence of 29 genes that were identified in tumors with more than one histotype, most gCIS identified on the basis of tumor histology were found in only one cohort (Fig. 4b, Supplementary Fig. 7 and Supplementary Data 1). Indeed, tumor histology was influenced by the same genes when engineered as initiating events or when altered through SB-mediated insertional mutagenesis (Fig. 4a, Supplementary Data 1, Supplementary Fig. 7). For example, papillary tumors were common in *Notch1*$^{ICD}$ and *Pik3ca*$^{(E545K \text{ or } H1047R)}$ cohorts (Fig. 4a and Supplementary Fig. 7). At the same time, both pathways were reciprocally activated by SB-mediated insertional mutagenesis in papillary tumors (Fig. 4a and Supplementary Fig. 7): either through activation of *Notch1* in R26-*Pik3ca*$^{(E545K \text{ or } H1047R)}$ tumors, or through loss-of-function insertions in *Pten*/gain-of-function insertions in *Eras* or *Igf2* in R26-*Notch1*$^{ICD}$ tumors (Fig. 4a and Supplementary Fig. 7). Thus, gCIS identified on the basis of histology-specific cohort analysis likely promote the histology in question (see below). Pathway enrichment analysis identified a number of the same signaling pathways or processes across multiple histological cohorts. For example, GTPase/Ras-mediated signal transduction, Wnt signaling (with different genes selected in tumors of different histotypes), protein complex assembly and polymerization, programmed cell death, microtubule polymerization, angiogenesis, neurogenesis, protein import/localization, protein catabolism/ubiquitination, and cytoskeletal organization were altered through SB-mediated mutagenesis in tumors of multiple histotypes (Fig. 5 and Supplementary Data 4).

Wnt and Notch signaling pathways coordinate development of many tissues[34]. Indeed, both have been implicated in mammary gland development and transformation of mammary epithelium[35,36]. While Wnt signaling regulates basal cell fate specification and mammary stem/bi-potent progenitor cell maintenance[37], Notch plays an important role in luminal progenitor maintenance/luminal cell differentiation[38]. As discussed above, many papillary tumors induced by hyperactivated *Notch1* selected for gCIS that were predicted to activate PI3K/AKT signaling. To validate this result and to test for the role of Notch in mammary tumor cell differentiation, we crossed R26-*Notch1*$^{ICD}$ transgenic mice to our *Pik3ca* mutant lines (E545K and H1047R). Indeed, *Notch1*$^{ICD}$ cooperated with both *Pik3ca* alleles to significantly reduce tumor latency and increase tumor number per mouse (Fig. 6a). In addition, virtually every tumor that formed in double transgenics showed papillary histology (Fig. 6a). Since SB insertions appeared to disrupt negative regulators of Wnt signaling in many *Pik3ca*-mutant mammary tumors, we also tested for cooperation between these pathways. To this end, we crossed mice with a Cre-conditional activated allele of *Ctnnb1* (*Ctnnb1*$^{\delta ex3}$) (Supplementary Table 1), the gene coding for β-catenin, with both *Pik3ca* mutant lines. As with *Notch1*$^{ICD}$, *Ctnnb1*$^{\delta ex3}$ cooperated with *Pik3ca*$^{E545K}$ and with *Pik3ca*$^{H1047R}$, except in this case induced tumors had squamous histology (Fig. 6b). These results are consistent with cooperative interaction between Wnt signaling and *Pten* loss as previously described[39]. Most significantly, while validating results from our screen, these data also suggest a model whereby Notch and Wnt signaling play an opposing role in defining tumor histology (papillary/luminal for Notch and squamous/basal for Wnt).

To test the idea that Notch and Wnt function antagonistically to dictate mammary tumor histology, we sought to reduce Notch1 signaling in the context of PI3K-induced mammary tumor formation. Specifically, we tested for the effect of deleting *Notch1*, a tumor suppressor in some tissues[40], on *Pik3ca*$^{E545K}$ and *Pik3ca*$^{H1047R}$-induced mammary tumor formation. Indeed, as with PI3K + Wnt, the majority of mammary tumors that formed in *Pik3ca*$^{gain-of-function}$/*Notch1*$^{loss-of-function}$ mice were squamous

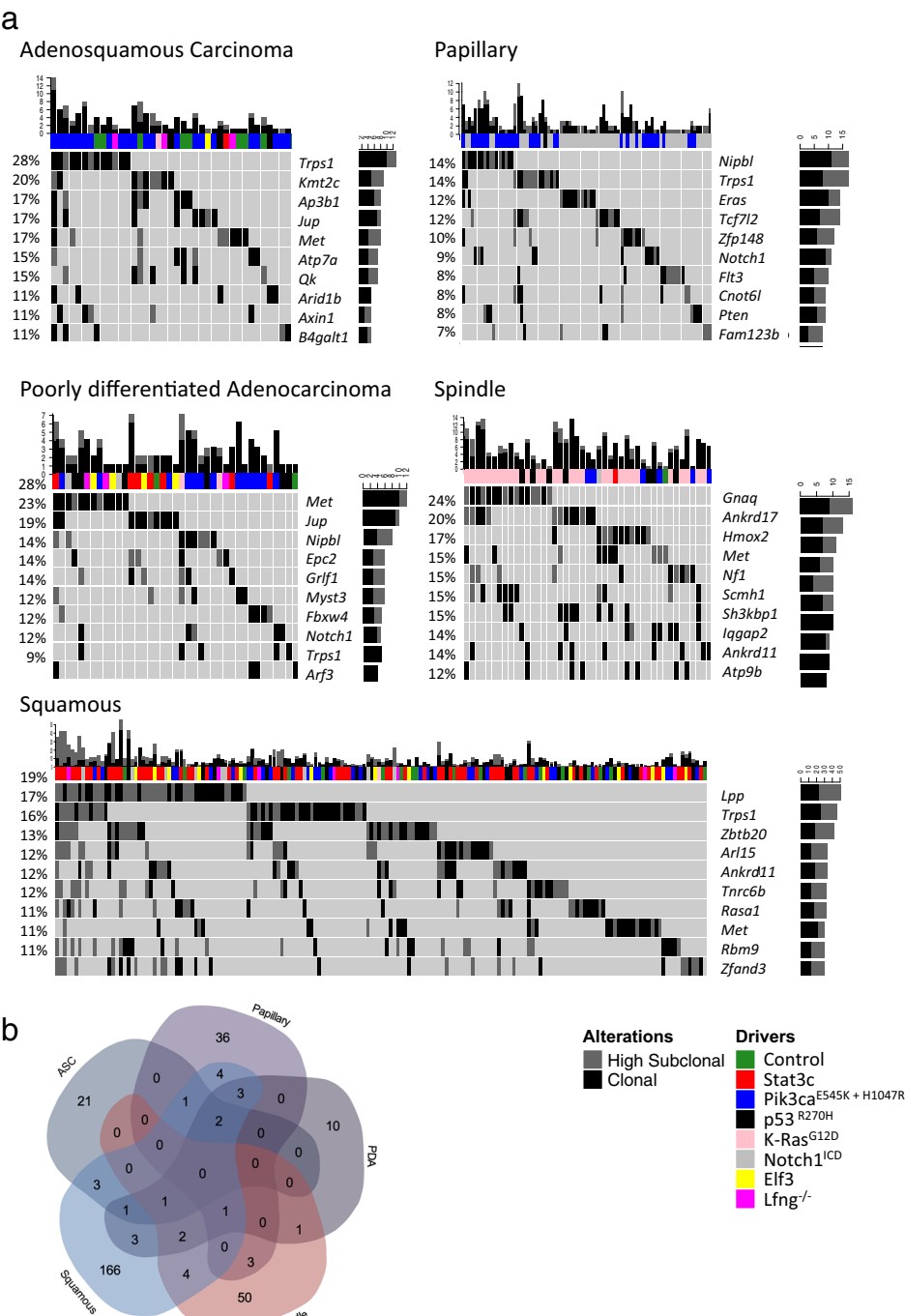

**Fig. 4 SB insertion site analysis on the basis of histology defined cohorts. a** Top gCIS identified in histology-based cohorts. The genetic background of each tumor is color-coded as indicated. Top 10 gCIS are shown for each. Note, the percentage of tumors with each gene targeted by SB is shown on the y-axis to the left for each bar graph, the number of tumors with clonal vs. subclonal SB targeting by SB is shown on the y-axis to the right, whereas the number of identified gCIS in each tumor is shown on the x-axis above each bar graph. Note, control tumors (in green) are those which developed in SB alone mice without a genetically engineered initiating event. **b** Venn diagram depicting the overlap of clonal gCISs identified in each histopathologic family.

(either adenosquamous carcinomas, squamous cysts, or squamous cell carcinomas). While *Pik3ca$^{H1047R}$* cooperated with *Notch1$^{loxP/loxP}$* to decrease tumor latency, *Pik3ca$^{E545K}$* did not (Fig. 6c). Thus, in mammary epithelium, *Notch1* can function as an oncogene in cooperation with *Pik3ca$^{E545K}$* and *Pik3ca$^{H1047R}$*, but as an allele-specific tumor suppressor in cooperation with *Pik3ca$^{H1047R}$*. The reason for allele specificity with respect to *Notch1* tumor suppressor gene function in this context is unclear. However, the lack of cooperation between *Notch1* gene loss and

expression of *Pik3ca$^{E545K}$* does not appear to affect the ability of *Notch1* deletion to skew tumors towards a squamous fate, revealing a separation between the ability of alterations in the Wnt/Notch signaling axis to promote transformation from their ability to effect tumor histology. In addition, the striking difference between *Pik3ca$^{E545K}$* and *Pik3ca$^{H1047R}$* in this assay is consistent with the very different list of gCIS identified in our SB screen for each allele (of 32 *Pik3ca$^{E545K}$*-derived and 62 *Pik3ca$^{H1047R}$*-derived gCIS, only 6 were identified in both screens

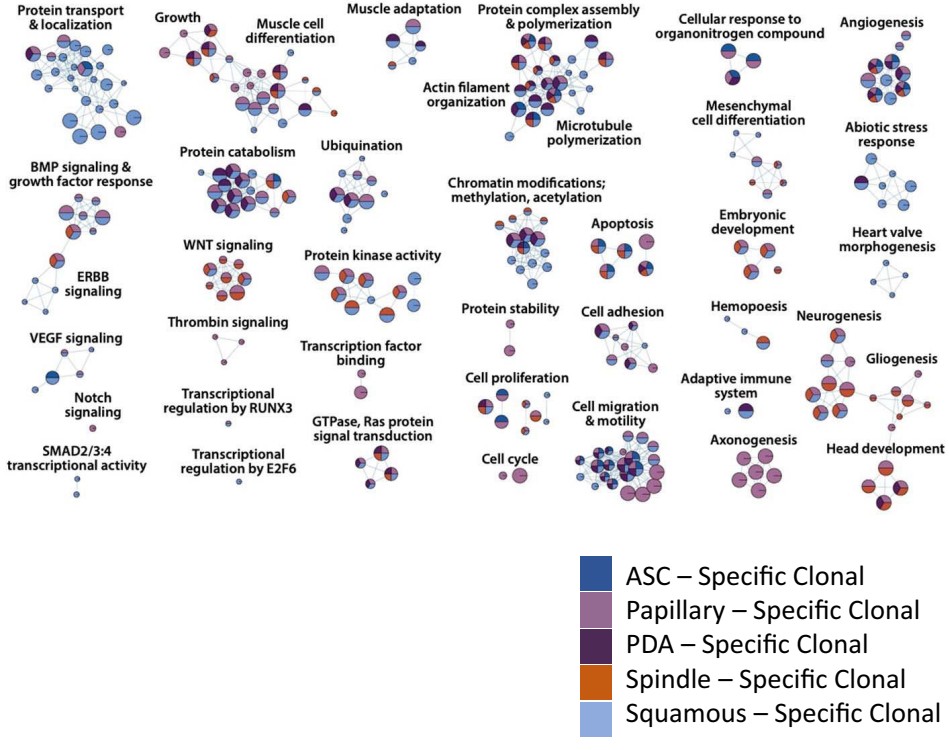

**Fig. 5 Pathway analysis identifies histology-specific pathways and processes in transformation of mammary epithelium.** Pathways altered by SB mutagenesis in histology-specific subscreens (FDR corrected *P*-value < 0.05). Enrichment map shows enriched pathways and processes as nodes, while pathways with many shared genes are connected and cluster into subnetworks. Node colors indicate cohorts where gCIS events were enriched in the corresponding pathways. One-sided ranked hypergeometric tests were used and significant pathways visualized. *p*-value information for nodes are found in Supplementary Data 4.

(*Met*, *Jup*, *Axin1*, *Myst3/Kat6a, Rab1*, and *Tm9sf3*) (Supplementary Fig. 3, Supplementary Data 1 and 5). Thus, our results are consistent with the notion that elevated Notch1 signaling drives tumor cell differentiation towards the luminal fate, whereas decreased Notch1 signaling or elevated Wnt signaling promotes tumor differentiation towards a more basal, skin, or pluripotent fate.

**Conditional haploinsufficient tumor suppressor genes show cooperative interaction**. As noted above, for the majority of gCIS identified, insertions were spread throughout the target gene and found in both orientations, suggesting loss-of-function. To directly test for changes in gCIS expression as a consequence of transposon insertion, we used NanoString technology to analyze RNA expression from *Pik3ca*$^{H1047R}$-SB mammary tumors[41]. Specifically, we tested for reduced or enhanced expression of all 62 *Pik3ca*$^{H1047R}$-cohort derived gCIS in RNA from 91 *Pik3ca*$^{H1047R}$-SB tumor samples. In this way, a large number of tumor samples without insertions in a specific gCIS could be used as negative controls for gCIS expression in tumors with insertions. A significant difference between samples with and without SB-insertions was seen for only 3 of 62 gCIS: *Arhgap8*, *Bcl11a*, and *Rai1*. Relatively modest overexpression was seen for two of three: *Arhgap8* (1.38-fold elevation, *p* = 0.0007) and *Rai1* (1.08-fold elevation, *p* = 0.045). In contrast, Bcl11a expression was elevated by an average of 2.1-fold (*p* = 0.00001) in samples with SB insertions targeting this gene. This is consistent with the transforming effect of Bcl11a when overexpressed[42]. We next tested for a link between gCIS-expression and tumor histology. Indeed, by unsupervised clustering, our 62 gCIS effectively distinguished squamous from papillary tumors within the *Pik3ca*$^{H1047R}$ cohort

(Fig. 7). Thus, expression of gCIS is more related to tumor histotype than to the presence or absence of transposon integration when assessed on a cohort level, suggesting that many SB targets may function to control mammary cell fate specification/differentiation. On the surface, it seems somewhat surprising that gCIS expression does not correlate with the presence or absence of SB insertions within a gene. We would predict that single allele disruption should reduce expression by half in most cases, and thereby promote tumor formation through loss of haploinsufficient tumor suppressor gene function. We still favor this idea, but note that since expression of many gCIS is lineage dependent, the effect of SB-mediated gene disruption on expression may well be impossible to detect based on phenotypic heterogeneity with associated noise in gene expression. Alternatively, for some gCIS, a neighboring gene whose expression was not measured, may be the real driver. Indeed, 59 pairs (or triplets) of neighboring genes were identified as gCIS in our screens (Supplementary Data 6). Perhaps, in some cases, insertions in one gene helped to promote tumor formation indirectly through it's effect on expression of its neighbor.

Functional genomic screens with SB, unlike screens using viruses like MMTV, involve mobilization of tens to hundreds of insertional mutagens (transposons) within each cell. As a result, this system has the potential to uncover or identify combinations of hemizygous loss-of-function mutations that cooperate in transformation. To test for this, we identified pairs of gCIS that showed higher-than-expected frequencies of co-occurrence. Such combinations conceivably represent cooperating genetic events within the same tumor cell, or events that cooperate with the driver in distinct cells (subclones) within the tumor. One pair of gCIS that co-occurred at a higher than expected frequency was *Nf1* and *Trps1*, which were targeted by presumed loss-of-function

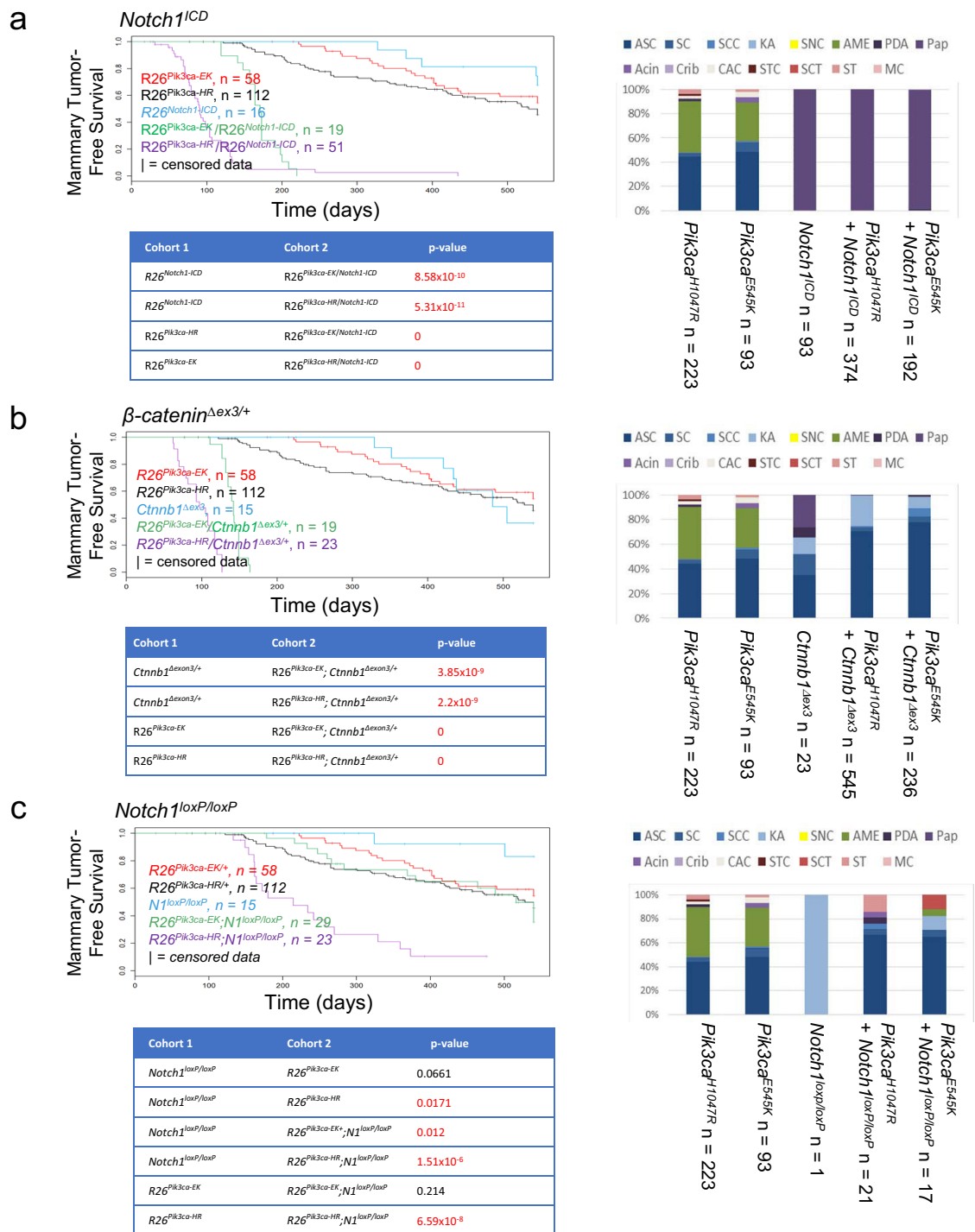

**Fig. 6 Mutant *Pik3ca* alleles cooperate with Notch and Wnt pathway alterations to promote histotype-specific mammary tumor formation in mice.**
Kaplan–Meier survival analysis (left) and distribution of histotypes for mammary lesions (right) from **a** mouse cohorts expressing *Pik3ca^E545K* or *H1047R* and activated *Notch1^ICD*, alone or in combination. **b** Mouse cohorts expressing *Pik3ca^E545K* or *H1047R* and activated β-Catenin, and **c** *Pik3ca^E545K* or *H1047R;Notch1^loxP/loxP* double mutant mice and controls. Mice in each cohort are also positive for MMTV-Cre^NLST. R26^Pik3ca-EK = Rosa26-LSL-Pik3ca^E545K mice, R26^Pik3ca-HR = Rosa26-LSL-Pik3ca^H1047R mice, R26^Notch1-ICD = R26-LSL-Notch1^ICD mice, β-catenin^Δexon3/+ = Ctnnb1^δex3/+ mice and N1^loxP/loxP = Notch1^loxP/loxP mice. For all Kaplan–Meier survival curves, statistical significance was calculated using the log-rank (Mantel-Cox) test.

insertions (clonal and subclonal) in 58 of 798 tumors (with identified gCIS) and 78 of 798, respectively. Fourteen of these tumors had insertions in both genes. This rate of co-occurrence is significantly higher than expected for random segregation of the specific oncogenic events ($p = 6.8 \times 10^{-4}$, two-tailed Fisher's Exact test). Three of the fourteen tumors with both genes

targeted were found in *Pik3ca^H1047R*-SB cohort mice. Therefore, to directly test for cooperation between hemizygous loss of both genes, we crossed mice with *Nf1* and *Trps1* mutant alleles (Supplementary Table 1) to *Pik3ca^H1047R* transgenic mice. While neither gene demonstrated haploinsufficient tumor suppressor activity on its own, they did when combined (Fig. 8). *Nf1*

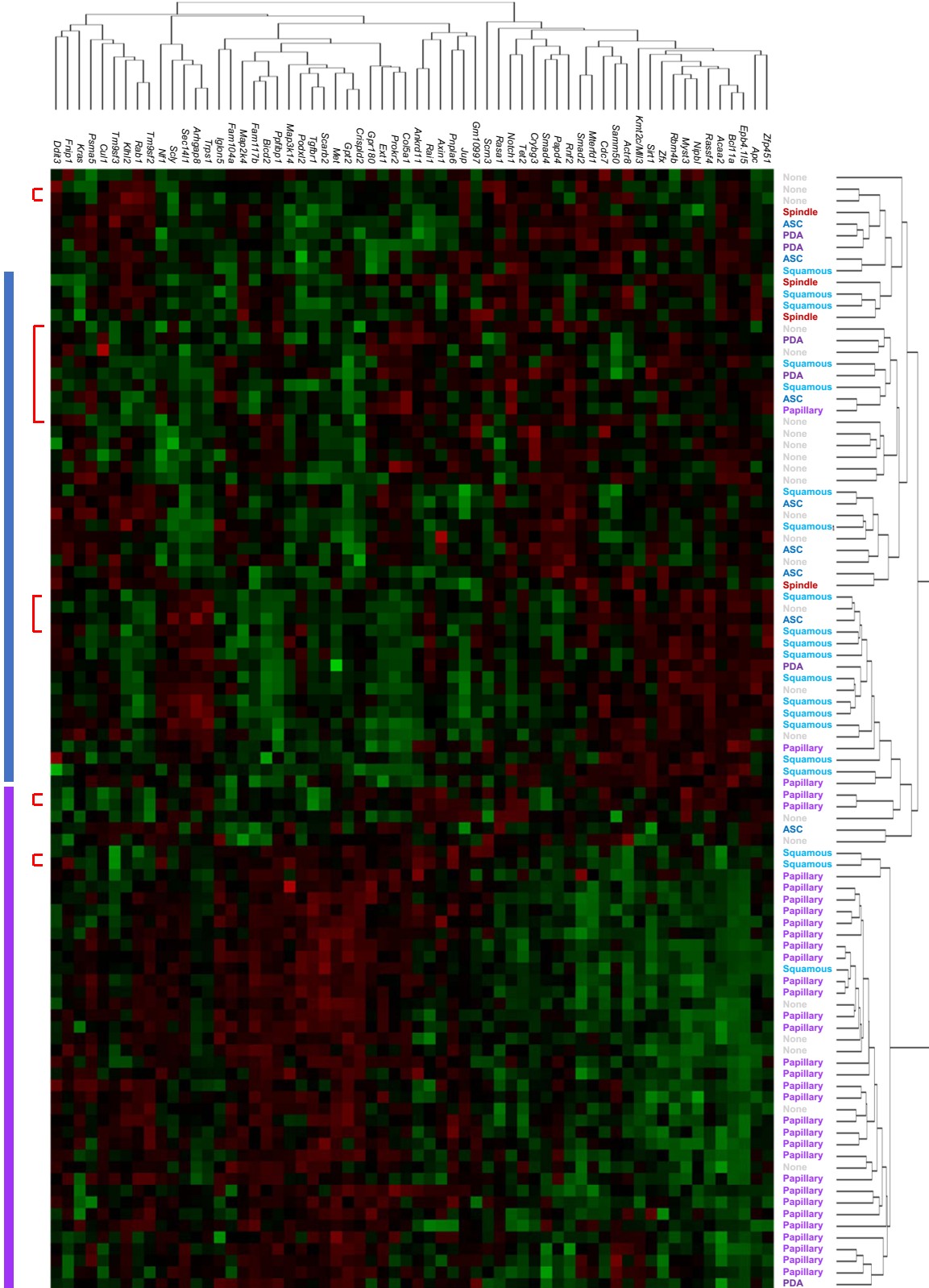

**Fig. 7 Hierarchical clustering of Nanostring gene expression data reveals histology-specific segregation of Rosa26-*Pik3ca*^H1047R SB tumors on the basis of gCIS expression.** Squamous and adenosquamous Rosa26-*Pik3ca*^H1047R SB tumors segregate separately from papillary and PDA tumors when clustered on the basis of gCIS expression. Red brackets show technical replicates using the same RNA sample.

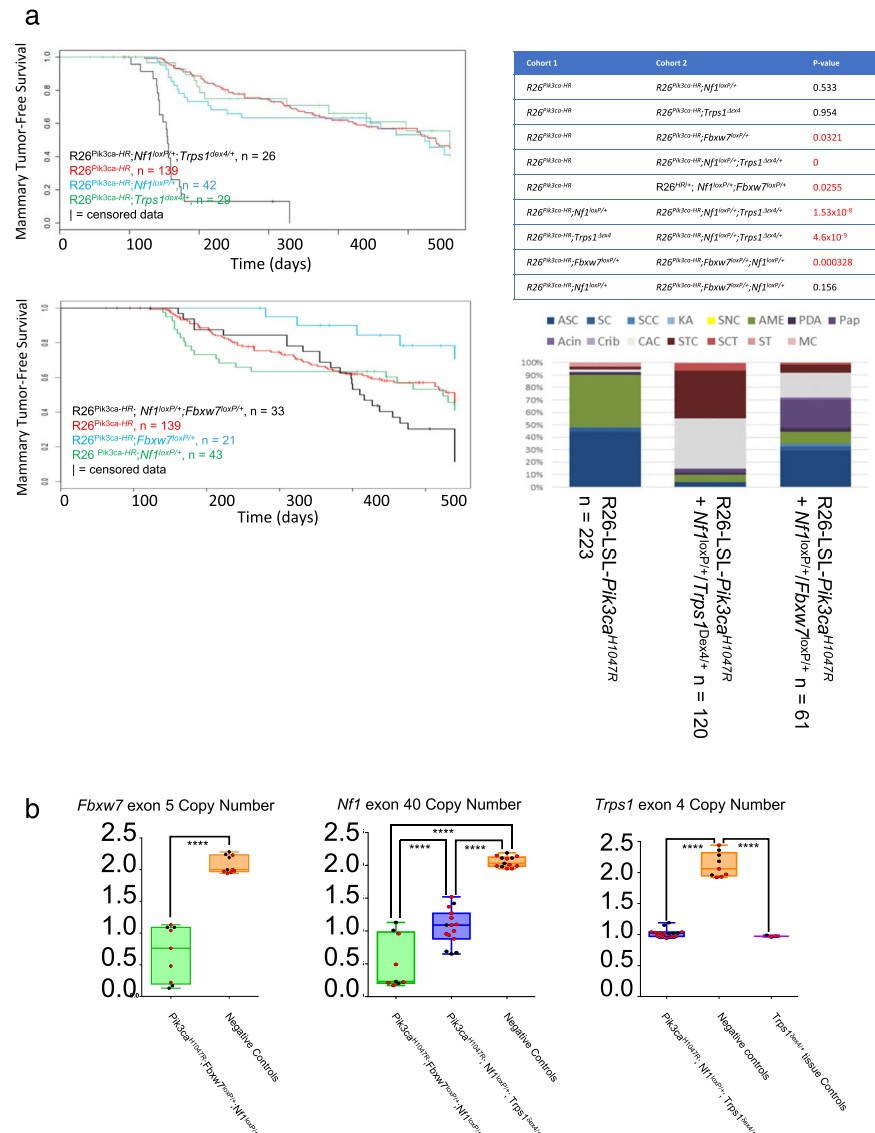

**Fig. 8 SB mutagenesis identifies conditional haploinsufficient tumor suppressor genes. a** Kaplan–Meier survival analysis for mice expressing *Pik3ca^H1047R* together with heterozygous mutations in *Nf1* (*Nf1^loxP/+*) and/or *Trps1* (*Trps1^δex4/+*), or neither gene (top). Kaplan–Meier survival analysis for mice expressing *Pik3ca^H1047R* together with heterozygous mutations in *Nf1* (*Nf1^loxP/+*) and/or *Fbxw7* (*Fbxw7^loxP/+*), or neither gene (middle). Bar graphs show histology of tumors that form in *Pik3ca^H1047R* or combined cohort mice. Mice in each cohort are also positive for MMTV-Cre^NLST. R26^Pik3ca-HR = Rosa26-LSL-*Pik3ca^H1047R* mice. For all Kaplan–Meier survival curves, statistical significance was calculated using the log-rank (Mantel–Cox) test. **b** ddPCR analysis shows *Fbxw7*, *Nf1*, and *Trps1* copy number analysis in mammary tumors from indicated genotypes. ddPCR assays were centered on the engineered mutation for each respective gene. This analysis reveals selection for *Nf1* copy number loss during selection of tumors in Rosa26-LSL-*Pik3ca^H1047R;Nf1^loxP/+;Fbxw7^loxP/+* but not in Rosa26-LSL-*Pik3ca^H1047R;Nf1^fl/+;Trps1^δex4/+* mice. Red data point are from lineage depleted tumor cells, which show less stromal contamination. Black dots are for whole tumor samples. Black and red dots are from distinct tumors. Statistical analysis was performed in R. For *Trps1* and *Nf1* data, a one-way ANOVA test with a Tukey's multiple comparison test was used, For *Fbxw7* data, a Welch's two sample *t* test (two-sided) was used. For each box and whisper plot, the center line represents the median (For *Fbxw7*: 0.76 (in *Pik3ca^H1047R;Fbxw7^loxP/+;Nf1^loxP/+* tumors) and 2.0 (in Negative controls)) (For *Nf1*: 0.23 (in *Pik3ca^H1047R;Fbxw7^loxP/+;Nf1^loxP/+* tumors), 1.09 (in *Pik3ca^H1047R;Nf1^loxP/+;Trps1^δex4/+* tumors) and 2.03 (in Negative controls)) and (For *Trps1*: 1.02 (in *Pik3ca^H1047R;Nf1^loxP/+;Trps1^δex4/+* tumors), 2.06 (in Negative Controls) and 0.971 (in *Trps1^δex4/+* tissue controls)). Box limits represent (lower) 25th to (upper) 75th percentiles. Whiskers show the min and max values. No outliers were identified. For *Fbxw7* copy number data in *Pik3ca^H1047R;Fbxw7^loxP/+;Nf1^loxP/+* tumors vs. negative controls, *p* = 4.53e−06. For *Nf1* copy number data, *Pik3ca^H1047R;Fbxw7^loxP/+;Nf1^loxP/+* vs. *Pik3ca^H1047R;Nf1^loxP/+;Trps1^δex4/+* tumors, *p* = 8.05e−05. For *Pik3ca^H1047R;Nf1^loxP/+;Trps1^δex4/+* tumors vs. negative control samples, *p* = 4.53e−11. For *Pik3ca^H1047R;Fbxw7^loxP/+;Nf1^loxP/+* tumors vs. negative controls, *p* < 2.00e−16. Finally, for *Trps1* copy number data, *Pik3ca^H1047R; Nf1^loxP/+; Trps1^δex4/+* tumors vs. negative control samples, *p* = 2.02e−14 and for *Pik3ca^H1047R; Nf1^loxP/+; Trps1^δex4/+* tumors vs. *Trps1^δex4/+* tissue controls, *p* = 8.16e−01 (not significant). Fbxw7 copy For ddPCR, *N* = 15 and 9 biologically independent tumors from R26-*Pik3ca^H1047R;Nf1^loxP/+;Trps1^δex4/+* and R26-*Pik3ca^H1047R;Fbxw7^loxP/+;Nf1^loxP/+* mice, respectively. *N* = 10 biologically independent controls for *Fbxw7^+/+* data, 13 for *Nf1^+/+* data, 9 for *Trps1^+/+* data and *N* = 3 for *Trps1^δex4/+*.

insertions were also co-selected with insertions in *Fbxw7* (58/798 tumors had *Nf1* insertions, 50/798 had *Fbxw7* insertions, 11 of which had both, $p = 4.7 \times 10^{-4}$, two-tailed Fisher's Exact test). Once again, a number of tumors with both genes targeted were found in *Pik3ca^{H1047R}*-SB cohort mice (4/11). We also saw cooperation between *Nf1* and *Fbxw7* hemizygous loss when combined in the context of *Pik3ca^{H1047R}* transgene expression, although this effect was much less significant (Fig. 8). Next, we used digital droplet PCR (ddPCR) to test for loss of the second copy of *Nf1*, *Fbxw7* and/or *Trps1* in tumors from the above-described cohorts and in tumorsphere cultures (with less stromal contamination), which would be expected if any of these genes were functioning as a recessive tumor suppressor in this context. Remarkably, one copy of *Nf1* was retained in tumors from R26-LSL-*Pik3ca^{H1047R}*;*Nf1^{loxP/+}*;*Trps1^{dex4/+}* mice, whereas significant loss of the second copy was seen in tumors from R26-LSL-*Pik3ca^{H1047R}*;*Nf1^{loxP/+}*;*Fbxw7^{loxP/+}*mice. *Fbxw7* also showed evidence for loss of the second copy in some R26-LSL-*Pik3ca^{H1047R}*;*Nf1^{loxP/+}*;*Fbxw7^{loxP/+}* tumors (Fig. 8b). In contrast, *Trps1* second copy loss was not seen (Fig. 8b). Thus, *Nf1* is a conditional haploinsufficient TSG gene (cooperating with activated *Pik3ca* on a *Trps1*+/− background, but not in the context of *Fbxw7* hemizygosity). Haploinsufficiency for *NF1* in general is likely very important in breast cancer since the hemizygous loss is much more common than homozygous gene disruption (~25% vs. ~1–3%, respectively)[4,13,14,25,43,44].

**Identification of candidate drivers behind human chromosome arm loss**. To put these data into a larger context, we compared our 1089 gCIS (clonal plus subclonal) with lists from retroviral and transposon-based cancer gene discovery screens in the mouse mammary gland. Interestingly, not many genes were shared across datasets (Supplementary Data 7). While 92 gCIS from our screen were identified as common targets in previous mammary gland SB screens, only two of these were also found in MMTV-based screens: *E-Ras* and *Jmjd1c*. Five gCIS in our screen (but not as clonal inserts in other mammary gland-specific SB screens) overlapped with those in MMTV screens: *Wnt1*, *Fgf3*, *Fgf8*, *Igf2*, and *Rreb1*. Two were found in other mammary-specific SB screens (but not in ours) and in MMTV screens: *Fgfr2* and *Nxn*. Interestingly, *Wnt1*, *Fgf3*, and *Fgf8*, known high-frequency MMTV targeted genes, were identified as subclonal gCIS in our screen.

Next we looked for overlap between genes identified in our screen and genes on the list of 99 focally-mutated oncogenes and tumor suppressors implicated in BC[2]. Importantly, four of our transgenic initiating genes were on this list (*PIK3CA*, *TP53*, *K-Ras*, and *NOTCH1*). Twenty-nine were identified as clonal and/or subclonal gCIS in our screen[2]. Interestingly, 17 of these were from the list of 241 gCIS identified as clonal in some cohorts and subclonal in others: *Axin1*, *Cbfb*, *Crebbp*, *Cux1*, *Fbxw7*, *Foxp1*, *H-Ras*, *Kdm6a*, *Kmt2c (Mll3)*, *Nf1*, *Notch1*, *N-Ras*, *Pik3r1*, *Pten*, *Smad4*, *Spen*, and *Usp9x*. Indeed, of the 351 clonal gCIS that were not found in any of our subclonal lists, only 7 were also found in the 99 human breast cancer gene list (*Apc*, *Arid1b*, *K-Ras*, *Mapk2k4*, *Map3k1*, *Nf2*, and *Tet2*). Only 5 of our exclusively subclonal gCIS (*Akt2*, *Braf*, *Cblb*, *Erbb2*, and *Ncor1*) were on the 99 gene list (Supplementary Data 7). Sixty-seven gCIS from our screen were found within a list of 299 human pan-cancer genes[45].

The relative lack of overlap is almost certainly related to selection for genes in our screen that promote mammary tumor formation when hemizygous. In contrast, the human BC gene list is mostly comprised of dominant oncogenes and recessive tumor suppressors. To put candidate hTSGs from our screen in context, we mapped their orthologous counterparts in the human genome.

Importantly, many map to chromosome arms showing hemizygous loss in approximately half of human breast tumors (e.g., 16q, 17p, and 8p (Supplementary Data 6). Specifically, we identified 22 orthologues of gCIS from our screen that mapped to 16q (*Orc6l*, *Gpt2*, *Lonp2*, *Siah1a*, *N4bp1*, *Cyld*, *Nudt21*, *Ciapin1*, *Coq9*, *Gpr114*, *Cnot1*, *Nae1*, *Cbfb*, *Tmco7*, *Cyb5b*, *Nfat5*, *Znrf1*, *Cmip*, *Crispld2*, *Zcchc14*, *Ankrd11*, *Gas8*), 15 that mapped to 17p (*Ywhae*, *Smg6*, *Srr*, *Pafah1b1*, *Spns2*, *Pld2*, *Rabep1*, *Wrap53*, *Ndel1*, *Myh1*, *Myh2*, *Map2k4*, *Ncor1*, *Cops3*, *Rai1*), and 12 to within regions of 8p that are commonly lost (*Tnks*, *Sgcz*, *Tusc3*, *Psd3*, *Slc18a1*, *Atp6v1b2*, *Xpo7*, *Chmp7*, *Ppp2r2a*, *Bnip3l*, *Kif13b*, *Rbpms*). While some cancer genes are tumor type-specific, others play a role in a broad range of cancers[45,46]. To increase the list of candidate hTSGs driving chromosome losses in multiple human cancers, we also mapped genes identified in SB screens from other tissues onto the human genome (Supplementary Data 6)[5,47].

**Many oncogenic mutations control signaling through initiating event-specific networks**. Heuristic literature-based analysis of gCIS identified in our screens revealed selection for mutations affecting either the initiating oncoprotein or processes/proteins regulated by this protein. For example, in *Pik3ca^{E545K}* cohort tumors, the most frequently mutated gene was *Fbxw7*, a known hTSG encoding a protein that is activated by PI3K signaling[48] and which promotes degradation of mTOR, a downstream signaling protein in this pathway[49]. Also, frequently targeted in these tumors is Cab39l, which regulates the Lkb1-Ampk-Tor pathway[50]. Similarly, *Ppp2r2a*, another gCIS selected in this cohort, codes for a regulatory subunit of PP2A that dephosphorylates phospho-Akt^{T308}. For *Pik3ca^{H1047R}* cohort tumors, the second most commonly targeted gene was *Kmt2c*, the disruption of which should decrease histone H3 lysine 4 methylation with altered transcriptional regulation of genes involved in proliferation[51]. This mutation should counteract Akt-mediated phosphorylation with resulting cytoplasmic sequestration of lysine demethylase 5A[52]. *Pik3ca^{H1047R}* tumors also selected for SB insertions in *Fnip1*, a tumor suppressor and regulator of mTOR activation at lysosomes[53]. *Fbxw7*, *Pten*, and *Eras* (with predicted gain-of-function insertions) were subclonal gCIS in *Pik3ca^{H1047R}* tumors, all of which should increase PI3K/mTOR signaling. Initiating event related gCIS were also selected for in *Notch1^{ICD}* and *Lfng^{loxP/loxP}* cohorts: *Fbxw7*, *Gsk3b*, and *Uxt* in the case of *Notch1^{ICD}*, and *Fbxw7* in the case of *Lfng*. As the *Elf3* screen was small, it is perhaps not surprising that we failed to identify known regulators, partners, or targets of Elf3.

Selection for insertions into genes affecting either the initiating oncoprotein or its direct downstream targets was obvious in *Trp53^{R270H}*, *K-Ras^{G12D}*, and *Stat3C* cohort tumors. In *Trp53^{R270H}* cohort tumors, insertions were identified in *Bach2*, *Cbfb*, *Cdk19*, *Ehmt1*, *Ep300*, *Kdm1a*, *Notch1*, *Phf2*, *Rybp*, *Trim 24*, and *Ubr5*. Each of these genes codes for a protein that directly regulates the transcription of *Trp53*, the stability or translation of *Trp53* mRNA, or the stability or activity of p53 protein (Fig. 9a). For *K-Ras^{G12D}* cohort tumors, insertions were identified in *Akap13*, *Dyrk1A*, *Dep1*, *Eif4enif1*, *Ints13*, *IqGap2*, *Met*, *Nf1*, *Nf2*, *Rasa1*, and *Sp3*: the protein products of which function to control mutant K-Ras protein activation (GTP loading), K-Ras-GTP availability for effector interaction, effector concentration or pathway feedback inhibition (Fig. 9b). Finally, in *Stat3C* cohort tumors, 24 different gCIS were identified that code for proteins regulating and/or sculpting Jak/Stat signaling (Fig. 9c). While further experimentation will be required to define precisely how each of these mutations function to promote/enhance initiating event-specific oncogenic transformation, working models can be developed. In the case of *Trp53^{R270H}*, it would appear that

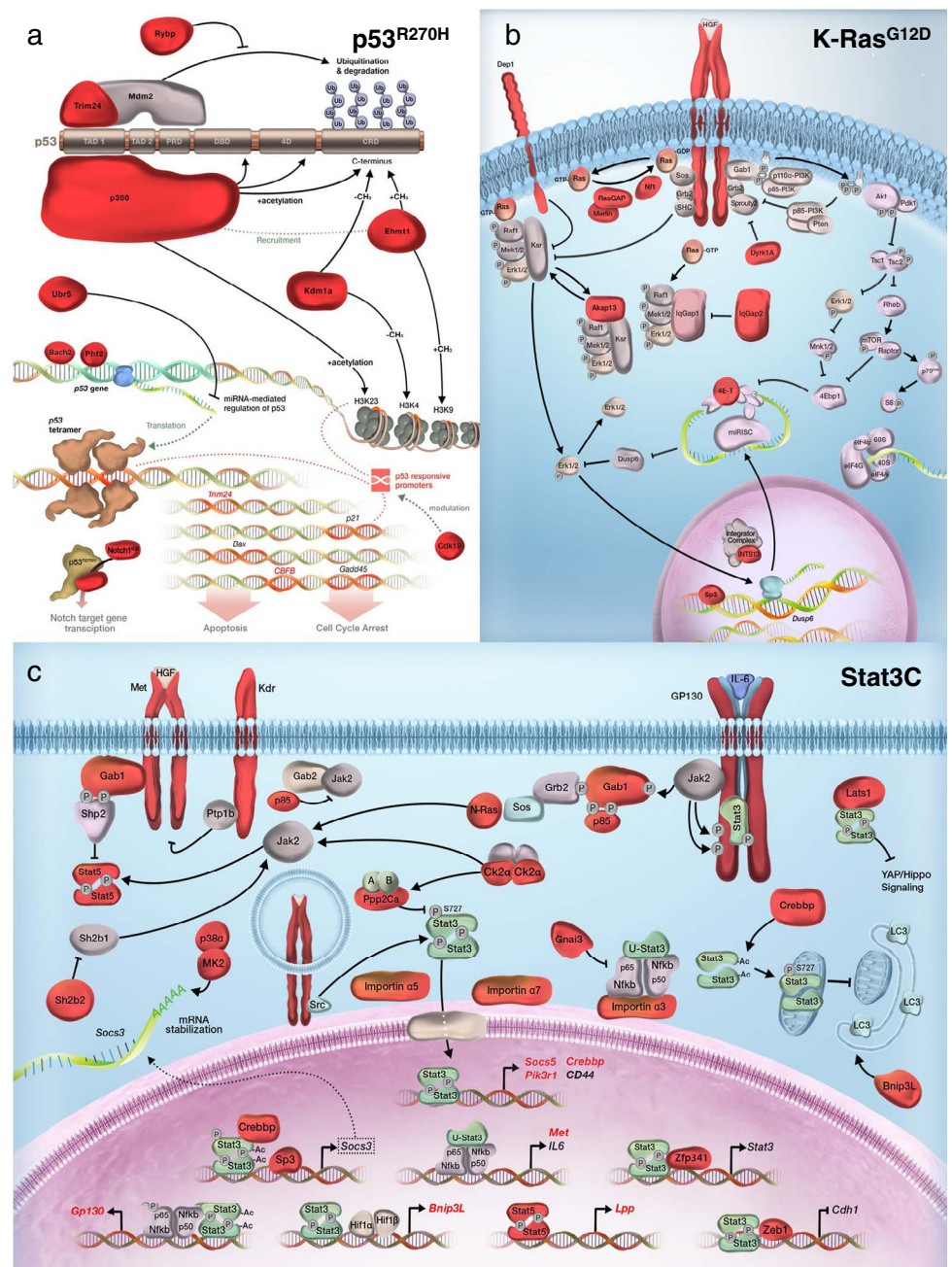

**Fig. 9 Many gCIS function on the same pathway as the oncogenic driver responsible for tumor initiation. a–c** Representative schematics highlighting molecular components of p53 (**a**), K-Ras (**b**), and Stat3 (**c**) pathways that were targeted for insertional mutagenesis in *Trp53^R270H^*, *K-Ras^G12D^*, and *Stat3C* cohort tumors, respectively. gCIS encoded proteins identified on each background are highlighted in red.

mutations that enhance mutant, rather than wildtype, function may be selected for[54]. In *K-Ras^G12D^* tumors, mutations that increase the level of available K-Ras-GTP, either through activation of GTP loading or loss of non-productive K-Ras-GTP:Gap protein interactions are being selected for, as are mutations that appear to promote the availability of specific effector complexes which include Ksr or IqGAP1. Finally, gCIS were identified that control or respond to Jak/Stat signaling in *Stat3C* cohort tumors. Since predicted hemizygous loss-of-function mutations were identified in genes that enhance Stat3-pY705 dimer formation (*Il6st*/gp130) or nuclear translocation (*Kpna1*/Importin α5 and *Kpna6*/Importin α7)[55]. perhaps potent transformation is associated with a specific ratio of tyrosine-phosphorylated Stat3 (and Stat5) dimers, Stat3 monomers (which

function together with NF-κB) and mitochondrial Stat3 complexes.

*PIK3CA* gain-of-function mutations occur together with copy number gains in ~7% of breast tumors[25]. In addition, two gain-of-function mutations occur within the same *PIK3CA* gene in 12–15% of cases[56]. Synergistic cooperation between such mutations has been demonstrated in cell lines and is associated with enhanced and sustained PI3K pathway signaling, as well as with increased cell proliferation and tumorigenicity of *PIK3CA*-virus infected cells[56,57]. To test for interaction between mutations on the PI3K pathway in vivo we studied tumor initiation and/or progression using an allelic series of *Pik3ca* mutants, each targeted to the Rosa26 locus on an FVB background. First, we tested for the effect of *Pik3ca^mutant^* gene dose using the same two

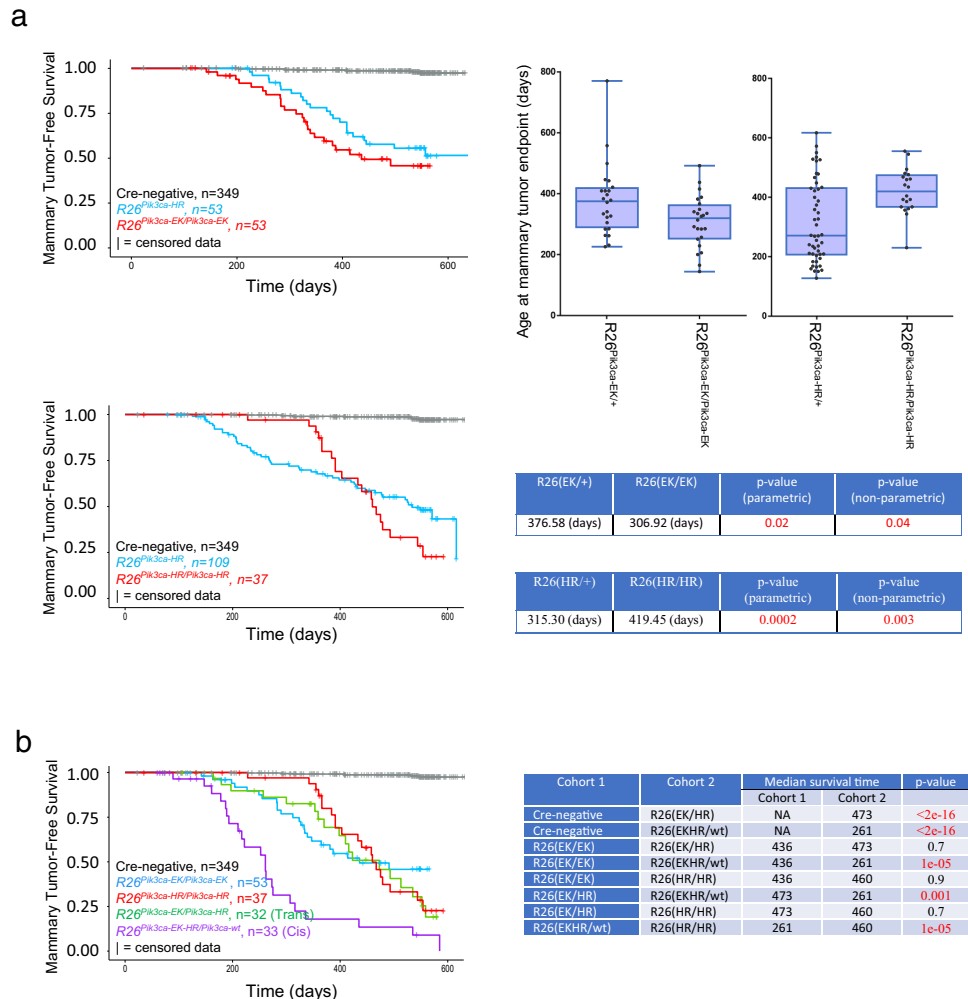

**Fig. 10 Genetic analysis of PI3K-induced mammary tumor formation defines cooperation between multiple pathway mutations.** Two copies of R26-LSL-*Pik3ca*$^{E545K}$ lower, while two copies of R26-LSL-*Pik3ca*$^{H1047R}$ raise, the age at mammary tumor endpoint (**a**). Kaplan–Meier survival analysis of *Pik3ca* hotspot mutations. Two-sided parametric *t*-test *p*-values were computed using the "*t* test" function in R. Two-sided non-parametric test *p*-values were computed using the "wilcox.test" function, also in R. Based on multiple testing ($n = 3$), *p*-values of less than 0.03 are considered significant. E545K and H1047R hotspot mutations show dramatic cooperation in cis in comparison to trans (**b**). Note, total *Pik3ca* transgene dose was controlled through ectopic expression of Pik3ca$^{wt}$ (by crossing in a R26-LSL-Pik3ca$^{wt}$ allele). Mice in each cohort were also positive for MMTV-Cre$^{NLST}$. For **b** two-sided *p*-values from the log-rank test comparing two survival curves were calculated using the "survdiff" function from the R/Bioconductor package "survival". For each box and whisper plot, the center line represents the median. Also, box limits represent (lower) 25th to (upper) 75th percentiles. Whiskers show the min and max values. There was one EK/+ outlier (770 days). No outliers were identified for other cohorts.

strains discussed above (R26-LSL-*Pik3ca*$^{E545K}$ and R26-LSL-*Pik3ca*$^{H1047R}$). Interestingly, while age at mammary tumor endpoint was significantly reduced with two copies of *Pik3ca*$^{E545K}$ (in comparison to one), the opposite effect was seen by doubling the dose of *Pik3ca*$^{H1047R}$ (Fig. 10a). Indeed, mammary tumor-free survival curves had a dramatically different shape when one or two copies of *Pik3ca*$^{H1047R}$ were present, suggesting a differential effect of increased gene dose on initiation vs progression or outgrowth (Fig. 10a). Next, when E545K and H1047R mutations were present on the same cDNA, survival time was dramatically reduced in comparison to the situation seen when both mutations were provided in trans (on different cDNAs) or when two copies of either mutation were expressed (Fig. 10b). Thus, tumor formation is significantly altered in the presence of multiple mutations on the driver pathway. This can mean selection for more signaling, as with increased *Pik3ca*$^{E545K}$ dosage or expression of *Pik3ca*$^{E545K-H1047R}$. However, this simple idea cannot easily explain the significant delay seen in mammary tumor endpoint in the presence of two rather than one copy of

*Pik3ca*$^{H1047R}$. These data, together with the surprisingly small overlap in gCIS identified as cooperating mutations in *Pik3ca*$^{E545K}$ vs. *Pik3ca*$^{H1047R}$ screens, the selection for non-classical hotspot mutations as second site allelic enhancers in BC, as well as evidence for unique signaling pathways downstream of each mutant[58,59], support a model whereby many breast tumors select for just enough and just right signaling within the PI3K signaling pathway(s).

## Discussion
Somatic copy number alterations affecting large chromosomal regions are very common,[4,5,60] with some occurring in more than 60% of BCs[4,60]. Within these segments, it is difficult to identify genes that when present at one copy or three copies (as opposed to two) contribute to tumor formation. To this end, functional genomic screens can be used. Historically, retroviral screens have led to the identification of very potent oncogenes in a number of tissues. More recently, Sleeping Beauty screens have helped identify putative drivers. Surprisingly, SB-derived cancer gene sets

are very different from those discovered in human cancer (which are biased towards genes that show recurrent focal alteration) or in viral screens. Naturally, these more traditional approaches to identify cancer genes would have missed those whose contribution is highly conditional or those which play a significant but modest transforming role in isolation.

Chromosome arm losses are difficult to study, since precise levels of gene suppression may have to be coupled to specific assays to identify transforming effects. Moreover, genes may show conditional haploinsufficiency that cannot easily be identified using one-gene-at-a-time analysis. In vivo RNAi screens as well as chromosome engineering approaches have begun to define important genes within large regions of recurrent chromosome loss[61–65]. Hemizygous loss of 8p, 17p, and 16q occur in ~50–60% of breast tumors. Our screen has identified dozens of candidate hTSGs that map to these regions alone. Other chromosome arms or large regions of the genome are lost in 15–30% of cases (e.g., 4p, 4q, 6q, 9p, 13q, 14q, 15q, 17q, 18q, 22q, and X). Many gCIS identified in our screen, most of which are candidate hTSGs, map to these regions as well. Some commonly deleted segments also contain known or suspected tumor suppressors. 16q, for example, includes genes for E-Cadherin (CDH1), Ubiquitin carboxyl-terminal hydrolase (CYLD) and Core binding factor β (CBFB). 17p includes TP53 and MAP2K4. 8p includes TNKS and BNIP3L. The most parsimonious explanation for these findings is that many recessive TSGs function as strong cancer drivers when both copies are lost. However, such genes may have conditional haploinsufficient tumor suppressor activity when combined with the appropriate set of other mutations. Indeed, Nf1 shows this property in the mammary gland (Fig. 8).

While TGFβ and BMP signaling are known to regulate proliferation, epithelial/mesenchymal transition, migration, and stem cell self-renewal in the mammary gland[66], the paucity of focal mutations in key components of these pathways could be taken to suggest that they do not play a central role in mammary tumor initiation/progression. Indeed, while the gene coding for Smad4, a transcription factor hub on both pathways, is a recognized BC gene, focal mutation of SMAD4 is relatively rare[2]. In our screens, however, predicted loss-of-function mutations were identified in Smad4, Smad2, Tgfbr1, Tgfbr3, Bmpr1a, and Acvr2a. As TGFβ and BMP pathways can function in opposition to each other, additional studies will be required to determine how mutations in these genes function to promote breast cancer, as well as in what cell-of-origin, in cooperation with which other oncogenic pathway(s), and towards the development of which BC histotype(s). Consistent with known haploinsufficient tumor suppressor activity for multiple genes on the TGFβ pathway[6], it is important to note that hemizygous loss is common for many of these genes in breast cancer (SMAD4—24%, SMAD2—22%, USP9X—21%, TGFBR1—19%, BMPR1A—18%, and ACVR2A— 18%)[25]. Indeed, SMAD2 and SMAD4 are closely linked on chromosome 18q21, and therefore, in most breast tumors where one gene shows hemizygous loss, the other is also reduced to hemizygosity. Another striking result involved selection for SB insertions within genes controlling splicing, alternative poly-A addition site selection and/or other aspects of RNA metabolism. For example, Wbp4 and Mbnl2, both on chromosome 13q in humans were targeted by SB in Stat3C tumors. Both genes show hemizygous loss in 25–35% of BCs[4,25], and this is associated with poor survival.

Our data suggest that distinct mammary tumor histologies are associated with selection for a mostly non-overlapping set of gCIS. In addition, Wnt-Notch signaling antagonistically defines tumor cell identity along a skin-basal-luminal cell fate axis. Indeed, Notch was both oncogenic and tumor suppressive in cooperation with elevated PI3K signaling, yielding a distinct tumor histotype in each case. Our findings are consistent with published literature on oncogenic signaling pathways and their role in defining histology[33,67]. Some mouse models, including several used here, involve hyperactivation of signaling pathways at a level that exceeds that seen in human BC. For example, the Notch1 allele in R26-Notch1ICD mice codes for an N-terminal truncation/ligand independent allele with a C-terminal deletion removing PEST sequences. The tumors that formed in these mice were mostly papillary (Fig. 4)[68]. In contrast, mutant alleles of Notch typically identified in human BC produce less active and ligand-dependent receptors[69]. These mutations are almost always seen in triple-negative tumors[69]. Thus, in contrast to the papillary tumors that form in R26-Notch1ICD mice, Notch mutants are associated with more basal-like breast tumors in humans. This is likely related to the fact that low-level Notch signaling may be sufficient to induce luminal progenitor self-renewal as required for basal-like BC, but not enough to force luminal lineage specification[70,71]. Also, Notch$^{\delta PEST}$ receptors would still require contact with ligand-expressing cells (e.g., basal-like cells)[72,73] for signal activation. Indeed, most mutations that activate Wnt or Notch signaling in human BC do so in a relatively subtle way. For example, low-level Notch activation occurs in 1q21-23 amplified BC as a result of overexpression of several genes on the Notch pathway[74].

Perhaps our most striking result was selection for cooperating mutations on the same pathways as the initiating event. This has been seen in large screens to identify mutations promoting gastrointestinal tumor formation. In that context, additional Wnt pathway mutations were selected for in ApcMin SB cohort lesions[31,75]. Kdm1A insertions, like those identified in our Trp53R270H cohort tumors, were also reported in a Trp53R172H screen from the same study[31]. Indeed, Pik3ca and Pten cooperation like that suggested by selection for Pten insertions in 10 Pik3caH1047R cohort tumors (Supplementary Data 1) has been shown in a model for ovarian cancer[76]. These data, together with findings in human cancer on selection for hypomorphs rather than null alleles in cooperation with germline APC mutants, have led to a model for selection of just right levels of oncogenic signaling that optimize tumor growth without engaging oncogene-induced senescence or other tumor suppressive systems[77]. This idea is also supported by selection for multiple mutations within the same oncogene or for selection for mutation plus copy number changes[56,78–80]. Moreover, multiple Ras pathway genes are mutated together in some human tumor types[81,82].

Our study highlights a unique aspect of the just right phenomena; selection for mutations that appear to sculpt the network that functions downstream of the main driver. Conserved signaling pathways like those associated with cancer interact with each other in complex ways. There isn't one PI3K, p53, Ras, or Stat3 pathway, but many. As Sleeping Beauty screens create a massive number of insertional mutations, it becomes possible to identify a host of haploinsufficiencies that enhance tumor growth. Indeed, our screens show strong evidence for loss-of-function insertion mutations that control signaling within a pathway or network. This was particularly clear for screens in Trp53R270H, K-RasG12D, and Stat3C, all of which involved cooperation between Sleeping Beauty transposon mobilization and expression of an oncogene from its endogenous promoter (Trp53R270H and K-RasG12D) or expression of a weak oncogene that was still dependent on growth factor/cytokine stimulation (Stat3C). Based on these data, we suggest that while individual tumor cells undergo Darwinian selection, the tumor as a whole appears to learn to regulate and sculpt initiating oncogenic signaling pathways, thereby establishing just enough and just right conditions to maximize tumor growth and progression. In contrast to hotspot mutations in oncogenes like PIK3CA, which are very potent, most

of the gCIS identified in our functional genomic screen likely contribute to mammary tumor formation in more subtle and conditional ways. Indeed, conditional cooperative haploinsufficient alleles, especially those linked within frequently lost chromosome arms, represent a powerful mechanism by which tumors can evolve to establish optimal growth conditions. This idea helps explain how mutations that disrupt genes with no obvious link to cancer, like those coding for importin α3, α5, or α7, can promote tumor formation or progression (Fig. 9). Ultimately, detailed knowledge of oncogenic pathway- or histotype-specific haploinsufficiencies greatly expands the concept of cancer genes to include a very large list of conditional drivers. Given the high frequency of chromosome arm level loss in BC, it seems likely that copy number changes among genes in this large list of conditional cancer genes function—in aggregate—as a major driver of tumor formation and/or progression. This more complete picture should help refine oncogenic pathway-specific therapeutic intervention.

## Methods

**Mouse colony maintenance and genotyping.** All mouse strains and their source are listed in Supplementary Table 1. These lines were maintained at The Centre for Phenogenomics (TCP) in accordance with guidelines established by the Canadian Council on Animal Care (CCAC). This study received ethical approval from the Animal Care Committee at the Centre for Phenogenomics. Only female mice were studied in mammary tumor experiments. Mouse breeding was initiated at sexual maturity. Experimental animals were virgins and tumor formation occurred as shown in Figs. 6, 8, 10 and Supplementary Fig. 2. Mice were genotyped with primer sets listed in the reagents and resources table (Supplementary Table 1).

**Necropsy and tumor collection.** Experimental mice were monitored for tumor formation over a period of ~540 days. When mice reached humane endpoint, they were sacrificed according to Canadian Council on Animal Care (CACC) guidelines. Upon sacrifice, mammary tumors were collected and a portion of each (along with adjacent normal mammary tissue) fixed in 10% buffered formalin phosphate (Fisher Scientific HC200-20) at room temperature for a minimum of 24 h. The remainder of each tumor was divided into smaller pieces and frozen on dry ice. Samples were placed at −80 °C for long-term storage.

**Histological analysis.** Formalin-fixed tissue samples were paraffin-embedded by the Pathology Core at the Centre for Modeling Human Disease (CMHD) in TCP. 5 μm sections were stained with Hematoxylin and Eosin, and used for histological analysis.

**SB insertion sequencing Shear-SPLINK.** Genomic DNA (gDNA) was extracted from snap-frozen SB mammary tumor samples (DNeasy Blood and Tissue Kit, Qiagen, Cat # 69506), and 5 μg of gDNA from each specimen diluted in ddH20 to a final volume of 100 μL. Samples were acoustically sheared to ~300 bp fragments with a Covaris S220/E220 Focused-Ultrasonicator (Covaris Inc., USA) using the following parameters: peak incident power (W)— 140, duty factor—10%, cycles per burst—200, treatment time—80 s, temperature 7 °C, water level—12 cm. Next, an Epicenter End repair kit (Lucigen Corporation, USA) was used with 20 μL of Sonicated DNA, 0.5 μL ddH₂0, 3 μL kit buffer, 3 μL dNTP, 3 μL ATP and 0.5 μL kit enzyme mix. Sample was incubated at RT for 45 min and then 10 min at 70 °C. Linker+ (5′-GTAATACGACTCACTATAGGGCTCCGCTTAAGGGAC-3′) and linker- (5′-Phos-GTCCCTTAAGCGGAG-C3spacer-3′) primers (100 μM) were mixed 1:1 in Sodium-Tris-EDTA buffer (50 mM NaCl, 10 mM Tris-Cl—pH 8.0, 1 mM EDTA—pH 8.0). Primer solution was heated to 95 °C for 5 min and slowly cooled to room temperature. Fast-link ligase kit (Lucigen Corporation, USA) was used with 30 μL end-repaired DNA, 1.75 μL ATP, 1.64 μl adaptor mix, 0.5 μL kit buffer, and 1.11 μL Fast-Link ligase. The resulting solution was then incubated at RT for 45 min and enzyme inactivated with incubation at 70 °C for 15 min. 35 μL of adaptor ligation solution from the previous step, 1 μL High Fidelity (HF) BamHI (New England Biolabs, Cat # R3136), 1.5 μL NEB buffer 4, 5 μL 10X bovine serum albumin (BSA), and 4 μL ddH₂0 were incubated overnight at 37 °C, column purified, and eluted in 50 μL EB Buffer (QIAquick PCR Purification Kit, Qiagen, Cat # 28106).

Two primary PCR reactions were set up for each side of SB transposons: (IRR and IRL). 5 μL DNA mix from the previous step was then mixed with 12.25 μL ddH₂0, 5 μL 5× Phusion buffer, 0.75 μL 10 mM MgCl₂, 0.5 μL 10 mM dNTPs, 0.5 μL 10 mM IRR (5′-GGATTAAATGTCAGGAATTGTGAAAA-3′) or IRL (5′-AAATTTGTGGAGTA GTTGAAAAACGA-3′) primer, 0.5 μL 10 mM Linker-A1 primer (5′-GTAATACGA CTCACTATAGGGC-3′) and 0.5 μL Phusion Taq (Sigma, USA). Each sample was then run using the following PCR cycle protocol: 1) 98 °C (30 s), 2) 98 °C (20 s), 3) 55 °C (30 s), 4) 72 °C (60 s), Steps 2,3,4 repeated 25 times, 5) 72 °C (60 s) and 6) 4 °C (hold).

3 μL of the primary PCR reaction was then diluted 1:50, vortexed and incubated at RT for 30 min. Secondary PCR mixes were made with 4 μL of DNA mix from the previous step plus 32.5 μL ddH₂0, 10 μL 5x Phusion buffer, 1 μL 10 mM dNTPs, 2 μL 2.5 μM IR-barcoded transposon primer (5′-AATGATACGGCGACCACCGAGATCTACACTC TTTCCCTACACGACGCTCTTCCGATCT(barcode)TGTATGTAAACTTCCGACT TCAACTG-3′), 0.25 μL 10 μM Linker-A2 primer (5′-CAAGCAGAAGACGGCATA CGAGATCGGTCTCGGCATTCCTGCTGAACCGCTCTTCCGATCTTAGGGCTC CGCTTAAGGGAC-3′) and 1 μL Phusion Taq. Touch down PCR cycling protocol was used: (1) 98 °C (180 s), (2) 95 °C (30 s), (3) 49 °C (30 s), (4) 72 °C (60 s), Steps 2,3,4 repeated 10 times, (5) 95 °C (30 s), (6) 53.3 °C (60 s) and (7) 72 °C (120 s). Steps 5, 6, 7 were repeated 25 times, (8) 72 °C (60 s) and 9) 4 °C (hold). PCR products run on the same lane were then pooled and purified using Qiagen purification kit and resuspended in 50 μL TE buffer. A Nanodrop was used to determine concentration of purified DNA. A maximum of 96 samples were pooled together from the IRL and IRR libraries per lane with a final concentration of 20–25 ng/ μL. This pool was incubated at 40 °C for 30 min and submitted for sequencing on the Hiseq (Illumina, USA) paired-end 2 × 126 bp.

**SB read preprocessing, alignment and analysis.** Adaptors were trimmed via cutadapt (v1.8) with parameters "-m 5 --no-indels --discard-untrimmed -g R1_5prime=^NNNNNNNNNTGTATGTAAACTTCCGACTTCAACTG" from read 1 (R1) for each sample. Since SB transposons recognize and insert into TA dinucleotide sequences, only reads starting with a TA were kept for downstream processing. R1 reads were then linked to their respective paired reads (R2) and aligned with novoalign (v3.05.01) using parameters "-r ALL 1 -R 0 -c 8 -o SAM" using the mm9 mouse genome assembly. Aligned sam files were converted to bams for downstream analysis. Each integration address was annotated using the refFlat tables from the UCSC genome database. Using chromosomal address, the following information was extracted: [tumor ID], [gene name], [region of gene hit (e.g., intron, exon, and promoter)], [predicted effect of insertion on the expression of the gene], [number of reads on this insertion site within the sample], [orientation of the transposon relative to the gene]. Some insertion events were not annotated because they did not occur within or near a known gene. The IRL and IRR libraries were then merged. If an insertion was detected in both libraries (i.e transposon orientations) the read and higher read count was used in the merged file. A dynamic filter was used to categorize insertions as clonal or subclonal. For each library, three thresholds were calculated using the insertion data: (i) >95% percentile of reads under the negative binomial distribution, (ii) 1% of the most abundant insertion sites, (iii) 0.1% of the total reads. The most stringent value was the threshold for clonal insertions, the second-most was the threshold for the clonal/subclonal category. Gene centric common insertion site (gCIS) analysis[21] was run on each cohort using clonal and subclonal/clonal insertion data. This test was repeated for every gene and then p-values adjusted using a stringent Bonferroni group-wise correction. Corrected p-values <0.05 were considered significant. Known false positive gCIS, *Sfi1* and *En2*, were manually removed from our results, as were gCIS that were identified on the basis of less than three independent/distinct insertions. Such genes were typically identified on the basis of multiple tumors within a single mouse. Also note, some mice were found to have both concatamers (Tg12740/T2Onc3a and Tg12775/T2Onc3b). Therefore, to avoid the possibility of misidentifying local hopping events as gCIS, we filtered out all insertions that mapped to mouse chromosome 9 and 12.

**Statistical analysis of mammary tumor-free survival and differences in proportionality.** All statistical calculations were made using R software. To analyze differences in mammary tumor-free survival, Kaplan–Meier (KM) survival curves were generated using the survival library and survfit function. Survival statistics were then calculated as non-parametric log rank p-values for censored survival data using the survdiff function. During each study, mice that reached endpoint due to conditions unrelated to mammary tumor development (typically either lymphoma or thymoma) were censored. To identify proportion inequalities between cohorts, 1-sided tests were calculated within a 95% confidence interval, using 1 degree of freedom. This was also performed with R software.

**Generation of Rosa26-targeted Cre-inducible mouse models for ectopic expression of Elf3 or Pik3ca^E545K-H1047R.** To clone Elf3 coding sequences, RNA was isolated from W4 mouse embryonic stem cells (mESC) (RNeasy, Qiagen, Cat # 74104; QIAshredder, Qiagen, Cat # 79654) and reverse transcribed (Quantitect Reverse Transcription Kit, Qiagen, Cat # 205310). 200 ng of template cDNA was used to PCR-amplify Elf3 with a forward primer that added a 5′ HindIII site with an in-frame N-terminal flag tag (5′-CTCAAGCTTGCCACCATGGACTACAAG GACGACGACGATAAGATGGCTGCCACCTGTGAGA-3′) and a reverse primer (5′- CGCTCTAGATTAATTCCGACTCTCTCCAACCTC-3′) that added a 3′ XbaI site. The Flag-Elf3 cDNA was subloned into a pBGT shuttle vector, which was subsequently inserted into the pRosa26Pam1 targeting vector. The Flag-Elf3_pBGT_pRosa26PAm1 composite vector was purified [EndoFree Maxi kit, Qiagen, Cat # 12362] and linearized with MluI. Similarly, *Rosa26-LSL-Pik3ca^E545K-H1047R* double mutant mice were generated as described for *Rosa26-LSL-Pik3ca^wt*, *Rosa26-LSL-Pik3ca^E545K* and *Rosa26-LSL*-Pik3ca^H1047R mice [18,83]. The *Pik3ca* double mutant version of the gene was generated based on fragments from single mutant Rosa26

targeting vectors[18,83]. For targeting, W4 mESC were eletroporated with linearized targeting plasmid and placed under G418 selection for 7 days. Correctly targeted clones were identified by 5' junction PCR using a forward primer that binds within the Rosa26 promoter, upstream of 5' homology arm (5'-CGCCTAAAGAA-GAGGCTGTG-3') and a reverse primer within the splice acceptor of the targeting vector (5'-GAAAGACCGCGAAGAGTTTG-3'). Correctly targeted diploid clones were submitted for morula aggregation at the Transgenic Core in TCP, and high-percentage male chimaeras were bred with FVB females until germline transmission was achieved.

To demonstrate transgene expression in R26^LSL-Elf3;MMTV-Cre mice, RNA from lineage-depleted mammary cells was extracted (RNeasy, Qiagen, Cat#74104; QIAshredder, Qiagen, Cat#79654) and reverse-transcribed into cDNA (Quantitect Reverse Transcription Kit, Qiagen, Cat#205310). A forward primer (5'-ctaggtaggggatcgggactct-3') and reverse primer (5'-cttatcgtcgtcgtccttgtagtc-3') were then used to detect a 381 bp sequence specific to the Cre-induced Flag-tagged Elf3 transcript. We tested for Cre-inducible $Pik3ca^{E545K-H1047R}$ transgene expression prior to morula aggregation. Specifically, G418-resistant mESC clones were transiently transfected (Fugene 6, Roche 11 815 091 001) with either Cre or control expression vectors (pCAGGS-Cre-IRES-puro or pCAGGS-FlpE-IRES-puro). RNA was harvested (RNeasy, Qiagen 74104; QiaShredder, Qiagen 79654) and reverse-transcriptase PCR performed (Oligo(dT) Primer, Invitrogen 18418-012; SuperScript II, Invitrogen 18064-022) using a forward primer in R26 exon 1 (5'-CTAGGTAGGGGATCGGGACTCT-3') and a reverse primer binding within the mouse $Pik3ca$ cDNA (5'-AATTTCTCGATTGAGGATCTTTTCT-3'). Cre-inducible mESC lines were used for morula aggregation by the Transgenic Core at The Toronto Centre for Phenogenomics (TCP). High-percentage male chimeras were crossed to FVB females. For both transgenic lines ($Pik3ca^{E545K-H1047R}$ and $Elf3$), germline transmission was confirmed by genotyping F₁ offspring. Upon completion of the T2Onc3b Elf3 SB screen, the Elf3 transgenic line was inadvertently lost. Thus, it was unavailable for screening using the T2Onc3a concatamer.

**Western blot analysis.** Western blot analysis was performed on selected tumor samples with, or without, predicted gain-of-function N-terminal truncation SB insertions in Notch1 or Jup. Tumor samples were run on Novex Wedgewell 8–16% Tris-Glycine gradient gels, proteins transferred to nitrocellulose membranes. Membranes were then blocked in TTBS with 5% Skim milk (Bioshop cat#SKI400) for two hours and probed overnight in TTBS/1% skim milk with a 1:500 dilution of Rabbit anti-Notch1 mAb (Cell Signaling D1E11 XP, catalog #3608) or with a 1:500 dilution of Mouse anti-Plakoglobin antibody (SCBT A6, catalog # sc-514115). Membranes were then washed in TTBS/1% skim milk, incubated with secondary antibodies (either 1:5000 anti-mouse IgG, HRP-linked antibody (Cell signaling cat #7076S) or anti-rabbit IgG, HRP-linked antibody (Cell signaling cat #7074S)). Finally, blots were washed in TTBS/1% skim milk, incubated in Pierce ECL Western substrate (Thermofisher cat#32209) and signal visualized on a Bio-Rad ChemiDoc system.

**Pathway enrichment analysis.** Pathway enrichment analysis was conducted according to standard protocols[29]. Significant genes in each group were ranked by $p$-value and analyzed in the g:Profiler web server [ref. 84, version e96_eg43_p13_3a389c1] using the ranked gene list option. High-confidence gene sets corresponding to biological processes of Gene Ontology and molecular pathways of the Reactome database were analyzed. Large gene sets (>1000 genes) and small gene sets (<5 genes) were filtered to improve interpretability of the analysis. The resulting significantly enriched pathways (g:Profiler default FDR < 0.05 (Benjamini-Hochberg) were visualized with the Enrichment Map software[85] in Cytoscape and resulting subnetworks of processes and pathways were manually annotated with the most prevalent functional themes (see Supplementary Table 1).

**Nanostring analysis.** Snap frozen tumor tissues were cut into ~2 mm³ pieces (about 15 mg) and used for RNA extraction with a Qiagen RNeasy plus mini kit (Cat #74136). In brief, tissue pieces were loaded into a tube contained a mixture of 1.4 mm (Qiagen cat. No. 13113-325) and 2.8 mm (Qiagen cat. No. 13114-325) ceramic beads in 350 μl of RLT plus buffer. Tubes were then placed on the Precellys 24 tissue homogenizer (Precellys EQ03119-200-RD000.0) and homogenized using 2 × 30 s at 5600 rpm with a 10 s break in between pulses. 320 μl of the obtained homogenate were then used for RNA extractions following instruction of the Qiagen Rneasy plus mini kit handbook. Nanodrop was used to check for purity and quantity of RNA.

Next, RNA samples were analyzed on a custom design Nanostring chip according to manufacturer's instructions in the Princess Margaret Genomics Centre at the University Health Network, Toronto. The N-solver analysis wizard was used on normalized data to perform agglomerative cluster analysis (with z-score genes and sample boxes checked, Spearman's Correlation and average linkage method settings used). Probe sets for $Pik3ca^{H1047R}$-SB gCIS are listed in Supplementary Data 8.

**ddPCR Analysis.** digital droplet PCR was used to determine copy number for mouse Fbxw7 exon 5, Nf1 exon 40 and Trps1 exon 4 using Mn00090804_cn (Chr 3:34967496 on GRCm38), Mn00351292_cn (Chr 11:79545398 GRCm38) and Mn00442178_cn (Chr 15:50664702 on GRCm38), respectively. One-way anova test was carried out using function aov(), Tukey's multiple comparison test was carried out using function TukeyHSD() and Welch's two sample $t$ test was carried out using t test() function in R.

**Reporting summary.** Further information on research design is available in the Nature Research Reporting Summary linked to this article.

## Data availability

Sequence data from Sleeping Beauty insertional mutagenesis libraries generated in this study have been deposited into the Gene Expression Omnibus (GEO) database under the accession code GSE143503. Source Data are provided with this paper and Sample annotation and barcodes are described therein. Source data are provided with this paper.

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

## Acknowledgements

We would like to thank Ramesh Shrivdasani, Joshua J. Paré, Ben Alman, Freddy Radtke, Tak Mak, Tim Lane, and Doug Melton for mutant and/or transgenic mice. The Egan lab acknowledges grant support from the Canadian Breast Cancer Foundation/Canadian Cancer Society Research Institute, the CDMRP/US Army Department of Defence, The Joint Canada-Israel Health Research Program with funding from the Canada's International Development Research Centre and the Canadian Institutes of Health Research. S.E.E., M.D.T., E.Z., and J.R.W. acknowledge funding from the Terry Fox Foundation. J.R. acknowledges funding from the Natural Sciences and Engineering Research Council of Canada (NSERC) Discovery Grant (#RGPIN-2016-06485). We acknowledge technical assistance and/or advice from Idil Temel, Toshi Kawamata, Judah Glogauer, Natalia M. Ruiz Agamez, Ruth G. Wong, Jack Plumaj, Nayasta A. Kusdaya, Alessandro C. Manno, Christine E.B. Jo, Jesse Joynt, Rameen Beroukhim, Lothar Hennighausen, Christine Watson, Robert Callahan, Robert Cardiff, Thomas Nalpathamkalam, Bhooma Thiruvahindrapuram, Jerid Robinson, and Liz Li. Finally, we wish to thank Life Science Editors (https://www.lifescienceeditors.com/), who assisted with writing and presentation of these results. We gratefully acknowledge Ben Pakuts for illustrations as shown in Fig. 9.

## Author contributions

Conception and design of the study: N.F.S., J.R.A., P.S., M.D.T., and S.E.E. Generation of Models: N.F.S., J.R.A., and A.J.D. Acquisition of data: N.F.S., J.R.A., P.S., K.J.K., C.A.L., N.R., J.Y., A.J.L., W.W., A.K., K.L.W., R.M.Q., D.D., J.L.G., D.W., J.S.S., P.L.G., G.P., A.J.D., L.G., and A.S.M. Analysis and interpretation of data: N.F.S., J.R.A., P.S., K.J.K., C.A.L., N.R., J.Y., A.J.L., Y.A., G.P., C.M.P., I.B.-P., R.K., E.Z., J.R.W., S.J.D., L.G., A.S.M., J.R., M.D.T., and S.E.E. Writing, review and/or revision of the manuscript: N.F.S., P.S., C.A.L., J.R., and S.E.E. Administrative, technical, or material support: A.J.L., G.P., and A.J.D. Supervision: J.R.W., J.R., M.D.T., and S.E.E.

## Competing interests

The authors declare no competing interests.
