## [Peer Review File · Nature Communications]

REVIEWER COMMENTS

Reviewer #1 (Remarks to the Author): Expert in mouse models of breast cancer

To the Authors,

In the manuscript entitled "Single allele loss-of-function mutations select and sculpt conditional cooperative networks in breast cancer" submitted to Nature Communications by Schachter et al., the authors perform a large-scale in vivo screening effort to identify gene aberrations potentially concurring with known breast cancer oncogenes. Therefore, the authors make use of the Sleeping Beauty transposon system in combination with conditional activation of tumor drivers in murine mammary tissue and sequencing of emerging tumors to identify loci recurrently targeted by the SB transposons. The comprehension of the screening setup of this study is impressive. However, the many findings of the screens lack to a large extent (i) validation on an expression level, (ii) functional validation, and (iii) validation of relevance for human breast cancer. Below, the authors can find detailed comments to the specific experiments and figure panels, many of which should be addressed to make this manuscript suitable for publication in Nature Communications.

Comment 1

In Supp. Fig. 1, the authors claim that SB mutagenesis, besides decreasing tumor latency, also increased penetrance. The authors should display tumor penetrance (e.g. via tumor counts and/or percentage tumor-affected glands) as separate data plots, as penetrance is rather difficult to be estimated from Kaplan-Meier curves.

Comment 2

In Supp Fig. 1, the SB transposon T2Onc3a and T2Onc3b donor concatemers apparently have a quite different latency in driving mammary tumors. The authors should clarify this difference, e.g. by looking at differential targeting of gCISs.

Comment 3

The authors identify a plethora of different gCISs, which they report in Fig. 1 and further throughout the study. For many, the insertion sites appear to be randomly spread across the respective loci, while for others, the insertion sites seem to cluster to specific regions of loci, such as the 5'-UTR and the first intron of Met. While the authors hypothesize and conclude the different effects of these transposonal insertions on gene expression (hemizygous loss of function and transcriptional activation, respectively), the study completely lacks evidence for these claims. It is crucial to proof hypothesized expressional changes (50% reduced expression, expression of truncated genes, etc.) using RNA and/or protein expression methods, such as Western blots or IHC. This is central to underline most claims and hypotheses throughout the manuscript.

Comment 4

Most of the gCISs identified appear to target genes randomly, presumably resulting in "single allele disruption". Because four of these genes are haploinsufficient tumor suppressors, the authors conclude that hemizygous loss of most of their genes results in "just enough and just right" oncogenic signaling pathway activation". However, the authors present no expression validation for these statements. Again, (i) if hemizygous loss of genes has a functional impact, then gene expression differences and (ii) consequently differential activation of the respective oncogenic pathways should be observed and described in the manuscript. (iii) Moreover, in human cancer, loss of heterozygosity (through genetic or epigenetic mechanisms) is a common phenomenon, which especially affects tumor suppressors, such as NF1 and PTEN. The authors should investigate whether LoH also takes place in their mouse models. If so, then their conclusion, hemizygous loss is "just enough and just right" for tumorigenesis, would be incorrect.

Comment 5

In Figs. 5 and 6, the authors confirm loss and/or activation of several of the genes identified in their SB screens to be involved in tumorigenesis. However, the manuscript completely lacks any proof of functionality of these different tumor models. The authors should at least genotype the emerging tumors to demonstrate switching of the conditional alleles. Moreover, it appears important to again show hemizygosity on a gene expression level (compare to comments 3, 4) as

much as protein expression of induced genes. The authors should demonstrate cytoplasmic/nuclear localization of active CTNNB1dEx5, reduced expression of Nf1, Pten, Fbxw7, as well as expression of truncated Trps1dEx4.

Comment 6

One of the important conclusions of this manuscript is the following: If gCISs are present at clonal levels, then the gene alterations are likely tumor-initiating. In contrast, gCISs present at sub-clonal levels are representative of tumor progression events. I agree with clonal equals tumor initiation, while I disagree with sub-clonal equals tumor progression. It is as (if not far more) likely that sub-clonal detection of gCISs is a caveat of bulk-sequencing of heterogeneous tumors, which emerged from multiple cells of origin driven by different gCISs. Although the authors convincingly identify (clonal) gCISs that co-occur in certain tumor subtypes and/or driver gene contexts, sub-clonality versus clonality likely just represents weaker versus stronger driver capacities in these respective contexts. Weaker (therefore slower) drivers are co-emerging with other driving events, thus appearing sub-clonal, while strong drivers outcompete weaker drivers and appear clonal. The efforts of the authors in Figs. 5 and 6 to confirm the tumor driving relevance of several of the gCIS hits via conditional inactivation of the gene (e.g. Nf1, Pten) are appreciated. However, unambiguous proof for relevance in tumor progression versus tumor initiation would require sequential induction of genes of interest (e.g. Pten loss) at a later timepoint (e.g. in established tumors) than the primary tumor driver (e.g. Pik3ca). Taken together, the authors should either rephrase their conclusions or demonstrate experimentally sub-clonal tumor progression versus multi-clonal tumors, the latter of which is challenging and might reach beyond the scope of this study.

Comment 7

Data related to the human relevance of found screen hits should be presented in a figure instead of as Supp. Table 5. The authors should present statistical evidence whether candidate genes identified in their SB screens are indeed enriched on chromosome arms frequently hemizygotously lost. The authors could also elaborate on this further, for instance by looking at expression of their screen hits in RNA-seq datasets of human mammary tumor cohorts. Along with this, the paragraphs describing comparison of screen hits to other transposon screens and to human breast oncogene and tumor suppressor genes should be either deleted from the manuscript (not very informative as table descriptions) or be presented as figure panels.

Comment 8

The different mouse strain abbreviations used are rather confusing. It would help the readability of the manuscript/figures, if gene names and gene variants were properly written out. Figure 2c and 4b have very limited value. It is unclear what all the dots and different colors mean. All the p-value tables are difficult to read, and should rather be presented as connecting lines on the plots itself. All the pie charts are difficult to read, especially in Figures 5-7. Shouldn't be referred to these by separate panel letters (a, b, c...)? In general, the Figures should be designed in a more appealing way.

Reviewer #2 (Remarks to the Author): Expert in mouse genetics and screens

In their study Schachter et al. performed Sleeping Beauty insertional mutagenesis screens in eight different engineered (breast cancer predisposing) genetic backgrounds. This strenuous effort produced more than 700 tumours and a list of over 1000 common insertion sites. Although several Sleeping Beauty screens have been performed earlier in breast cancer, this is the first study to perform such a large-scale systematic comparison in a large number of genetic backgrounds using the same Cre line. This approach allows direct in vivo interrogation of a series of questions, including genetic interactions, the genetic basis of phenotypic diversification, oncogene dosage effects and others. Generally, the study is well done, involving scientists with long-standing expertise in the design, execution and analysis of transposon screens. The data demonstrate the power of the screening technologies to uncover biological principles and will certainly be a very valuable source of knowledge in the field. However, I have major concerns about the presentation, the interpretation and discussion of the

data, which is in large parts confusing and requires substantial revision. In almost all sections of the manuscript, it is impossible to understand which of the conclusions/discoveries are new? In other words, the authors fail to convince the reader that the screens can uncover new biology, beyond what has been known before. As detailed below, I would suggest to clearly point out data that merely support known biology and to explain/present such data in a more condensed manner. Instead, the focus should be much more on the new discoveries, their interpretation, functional validation and discussion.

Specific comments:

A. Abstract

1. The abstract is a very poor summary of the scope, the results and conclusions of the study. In my eyes, it does not describe adequately the unique scale (700 cancers) of the study or the large number of biological questions that could in principle be interrogated (e.g. histological subtypes are missing). Instead, haploinsufficiency of Pten, Nf1, Fbxw7 and Trps1 is presented as a main discovery. For all four genes haploinsufficiency has been extensively reported before.
2. Main novel discoveries should be the guiding theme of the abstract
3. The "just enough and just right" concept of oncogenic signaling is put forward, without showing functional data. Although I support the interpretation, I feel that a more cautious wording would be more adequate to avoid overstatements (e.g. the data support the concept of "just enough and just right...")

B. Results section 1 (page 4):

1. Overview. The study uses with a series of genetic backgrounds and crosses. It would be helpful to have an overview figure/table containing the different genotypes/crosses, number of mice, number of tumors and related histologies, number of CISs identified.
2. In addition, because this manuscript represents a large resource, it would be useful to provide a supplementary table with detailed information for each individual mouse: (i) genotype, (ii) tumour number and histologies, number of insertions.
3. Transposon lines used and analyses. The authors state that most of the screens were performed using both the T2Onc3a und T2Onc3b transposon mouse lines (differing in the transposon donor chromosome). I could not find further information on this point. Were donor loci/chromosomes excluded from the analysis because of local hopping? How did the transposon line influence the data analyses.
4. Data analyses. Clonal vs. subclonal insertions: Although cutoffs for clonal and subclonal insertions are mentioned, the details remain unclear (but are important for an in depth understanding). How were the cutoffs defined? How many clonal/subclonal insertions did the authors find per tumor? How does the distribution look like? Were differences in clonality detected?

C. Results sections 2 (page 5) and 3 (page 7):

1. Supplementary figure 2 contains important information and should be presented in the main manuscript, by adding the plots to Figure 1.
2. These sections show extensive context-specificity of the CISs identified in the study: in different crosses/predisposing backgrounds CIS lists (genes and related pathways) differ substantially. The data impressively demonstrate the power of insertional mutagenesis to uncover biological principles. The authors list a series of examples of how the screen pinpoints for example cooperativity between genes and pathways. However, it remains unclear what is new and what has been known before (particularly for readers who are not experts in breast cancer biology). It is essential to rewrite these two chapters (but also many of the subsequent ones) in order to:
(i) discuss data related to known biology and associations in a more condensed form, and
(ii) point out new discoveries/biology, for which the authors should provide in depth functional experiments/data (at least for one example).
In addition, to help readers navigate through the the large and complex data sets, it would help to end each section of the manuscript with one or two sentences summarizing the main findings and putting them into context.
3. Figure 2c and 4b (pathway analyses). The figures display important findings, but are very hard to comprehend because (i) dots are too small and (ii) it seems that colours and labels have been mixed up. For example, in figure 2c Stat3c should be pink (see legend) but the dots in the pathways which are found in the Stat3c cohort (Pi3k or Wnt, see main text) are not pink. Likewise, Fgfr/Toll-like and IL signaling are exclusive for Stat3c according to the text but are colored dark

blue, which corresponds to PIK3CA HR according to the legend.

D. Results sections 4 (page 8) and 5 (page 9).

1. These two sections aim at establishing a link between genetics and histopathology. The data are important. I also like the presentation of data in figure 3a. However, the interpretation and discussion is confusing. It is again impossible to understand whether the authors were able to use the data in order to uncover new biology, beyond what has been known before. After reading the chapters multiple times, one remains confused. For example, the authors focus on Notch, PI3K and Wnt signaling for downstream analyses and validation. A Pubmed entry for breast cancer displays 800, 1900 and 3700 publications on Notch, Wnt or PI3K signaling, respectively. The link between papillary tumours and Notch1, for example, has been known before and it seems that the link between Wnt and the more basal subtypes is also common knowledge.

2. The authors cross Notch1-ICD and PI3K mutant mice. Based on this experiment they suggest that there is collaboration between Notch1-ICD and PI3K alleles in induction of papillary histology. However, the penetrance of papillary tumours is already 100% in the Notch1-ICD model.

3. Page 10: "Notch1 can function as an oncogene in cooperation with Pik3caE545K and Pic4caH1047R, but only as an allele-specific tumor suppressor in cooperation with the latter". This finding is interesting. What is the interpretation?

E. Results sections 6 (page 11) on cancer gene cooperation.

1. Cooperative haploinsufficient tumor suppressors: "We identified pairs of gCIS that showed higher-than-expected frequencies of co-occurrence" (p 11 line 9). Which other pairs besides Nf1-Trps1 and Nf1-Fbxw7 were detected? How many pairs were found to be statistically significant? Were these pairs found in combined analyses (across cohorts) or are there background specific pairs?

2. Figure 6, cooperation between monoallelic loss of Nf1-Trps1 and Nf1-Fbxw7. Does LOH occur in tumors? In addition, the authors might consider speculating about the biology of this collaboration in their discussion.

F. Results section 7 (page 12):

1. Minor comment: The authors might consider mentioning that potential confounders of analyses linking clonal/subclonal insertions to early/late tumorigenesis are not considered here, e.g. when an early driver insertion is not needed during later progression, transposons will jump out of the locus in many cells. Clonal sweep is another example.

G. Results section 8 (page 13).

1. Minor: please include the term "human" to avoid confusion: Identification of candidate drivers behind human chromosome loss.

A table, rather than long lists of genes in the main text might improve readability. More generally, the authors state that "many [identified hTSGs] map to chromosome arms showing hemizygous loss [in breast cancer]" (p13 last line). Although some examples are given, it remains unclear what "many" means. Is there a statistically significant enrichment in these regions compared to others?

H. Results section 9 (page 14).

1. To me this is the most fascinating part of the study: selection for insertions in genes affecting the pathway of the initiating engineered oncoprotein. Unfortunately, the authors do not show any functional data. However, they present in the previous section that Pten loss cooperates with PI3K. I would suggest to move those data into this section to strengthen this important aspect of the study.

2. Minor: The authors discuss the importance of oncogenic dosage variation in cancer only in a model in which an oncogene/pathway is co-regulated/modulated by other genes/pathways. They might consider mentioning another model, the dosage variation of the oncogene itself (e.g. PMID: 29364867, for example through copy number gain), which is a widespread phenomenon. It strengthens the point of the whole section as it provides compelling evidence for the validity of the "just enough and just right" concept that the authors put forward.

I. Further minor remarks:

In several figures, the authors might consider improving data presentation to increase accessibility (labeling, colors, style, quality, readability). In addition, there are a number of mistakes that need

to be corrected. For example:

1. Labeling of cohorts: using a consistent labeling of the cohorts would lead to an improved understanding (the Notch1 cohort is labeled as Notch1 ICD in Figure 1/2/4, as RN1 in Figure 3 and as N1-ICD in Figure 5. Comparable for p53 R270H which is sometimes only labeled as R270H).
2. Using the same colors for the cohorts in Figure 2 and 3 would make it more comprehensive
3. Figure 4a: For easier comparison of the histological subtypes between cohorts I would suggest a bar graph representation with only one legend.
4. Figure 4b: More colors in the dots than in legend. There are multiple yellow dots in the Figure but yellow is not described in the legend.
5. Legends of pathway analysis figures are very blurred and some dots too small to see different colors
6. Figure 6: Top and middle panel are not labelled correctly (change Figure legend).

Text and figure legends:

1. Why are the two cohorts of Pik3ca summarized as only one in Figure 3 although tumor spectrum seems to be different according to Figure 4?
2. Figure legend Figure 4: "Pathways altered by SB mutagenesis in GEMM-specific subscreens": copy and paste mistake from Figure 2c.
3. Figure legend Figure 5: three identical combinations of Kaplan-Meier curves and pie charts are labelled as:
 - a) Kaplan-Meier survival analysis (top) and histotypes of mammary lesions (bottom)
 - b) Survival analysis (left) and pie charts (bottom)
 - c) Survival analysis (top) and pie charts (bottom)
4. Introduction, page 3: "Although small scale SB screens have been performed..." The cited screens were mostly adequately sized for the questions that they addressed.

REVIEWER COMMENTS

Reviewer #1 (Remarks to the Author): Expert in breast cancer mouse models and screens

To the Authors,

In the manuscript entitled “Single allele loss-of-function mutations select and sculpt conditional cooperative networks in breast cancer” submitted to Nature Communications by Schachter et al., the authors perform a large-scale in vivo screening effort to identify gene aberrations potentially concurring with known breast cancer oncogenes. Therefore, the authors make use of the Sleeping Beauty transposon system in combination with conditional activation of tumor drivers in murine mammary tissue and sequencing of emerging tumors to identify loci recurrently targeted by the SB transposons. The comprehension of the screening setup of this study is impressive. However, the many findings of the screens lack to a large extent (i) validation on an expression level, (ii) functional validation, and (iii) validation of relevance for human breast cancer. Below, the authors can find detailed comments to the specific experiments and figure panels, many of which should be addressed to make this manuscript suitable for publication in Nature Communications. **The three issues noted here are addressed under specific comments below.**

Comment 1

In Supp. Fig. 1, the authors claim that SB mutagenesis, besides decreasing tumor latency, also increased penetrance. The authors should display tumor penetrance (e.g. via tumor counts and/or percentage tumor-affected glands) as separate data plots, as penetrance is rather difficult to be estimated from Kaplan-Meier curves.

We have now described the effect of SB on tumor formation in each cohort differently, and more conservatively. This is because the effect, as pointed out by reviewer 1, is complex. For example, the effect of Onc3a and Onc3b appears to be different (see below). This is almost certainly related to the fact that SB concatemers in Onc3a and Onc3b are on different chromosomes, and local hopping will hit different genes. Each screen has its own ratio of tumors from Onc3a vs Onc3b. In the original background, we stated “However, SB mutagenesis significantly reduced tumor latency on all genetic backgrounds (Supplementary Figure 1) and, in most cases, also increased tumor penetrance.” This has been changed to “However, SB mutagenesis either reduced tumor latency and/or increased incidence in most GEMM tested (Supplementary Fig. 2)(New).” Additional discussion of this point in the text of the manuscript would have the unintended consequence of suggesting that we can make conclusions based on changes in latency or penetrance (unfortunately, we can't).

Comment 2

In Supp Fig. 1, the SB transposon T2Onc3a and T2Onc3b donor concatemers apparently have a quite different latency in driving mammary tumors. The authors should clarify this difference, e.g. by looking at differential targeting of gCISs. **All gCIS identification have been performed in aggregate, where data from Onc3A and Onc3B screen have been combined. This increased the statistical power of our analysis substantially. It seems likely that the SB concatemer in Onc3A may have access to genes through local hopping which can enhance tumor formation to a larger extent than those that are easily targeted by Onc3B via local hopping. Either way, we remove donor chromosomes from our analysis to avoid artifacts and therefore do not have this data. We have redone the SB analysis in order to include Pik3ca^{E545K} in the large combined cohort. (as it was not included last time) but have not redone it by separating Onc3a from Onc3b cohorts. This is primarily because the statistical power drops significantly in many cohorts when we cut the sample number in half. Another confounding factor is that a small number of tumors showed evidence for SB mobilization from both concatemers. This must have resulted from a mistake in breeding, as T2Onc3a and Onc3b cohorts were kept separate. To account for this, and to avoid including false**

positives associated with local hopping, we removed all genes on chromosome 9 and 12, regardless of cohort (see statement in methods section on SB read preprocessing, alignment and analysis).

Comment 3

The authors identify a plethora of different gCISs, which they report in Fig. 1 and further throughout the study. For many, the insertion sites appear to be randomly spread across the respective loci, while for others, the insertion sites seem to cluster to specific regions of loci, such as the 5'-UTR and the first intron of *Met*. While the authors hypothesize and conclude the different effects of these transposonal insertions on gene expression (hemizygous loss of function and transcriptional activation, respectively), the study completely lacks evidence for these claims. It is crucial to proof hypothesized expressional changes (50% reduced expression, expression of truncated genes, etc.) using RNA and/or protein expression methods, such as Western blots or IHC. This is central to underline most claims and hypotheses throughout the manuscript. **This is an important point and we have performed a series of experiments to address it. First of all, to test for changes in gCIS expression as a consequence of transposon insertion, we used NanoString technology to analyze RNA expression from every *Pik3ca*^{H1047R}-SB mammary tumor sample available in the freezer. Specifically, we tested for reduced or enhanced expression of all 62 *Pik3ca*^{H1047R}-cohort derived gCIS in RNA from 91 *Pik3ca*^{H1047R}-SB tumor samples. In this way, a large number of tumor samples could be used as control for expression of each gCIS (i.e. tumors which are not driven by insertions in the same gene, represent excellent controls). A significant difference between samples with and without SB-insertions was only seen for 3 of 62 gCIS. Indeed, expression of gCIS was more related to tumor histology than to the presence or absence of transposon integration when assessed on a cohort level, suggesting that many SB targets may function to control mammary cell fate specification/differentiation. In short, this question cannot be addressed since there is so much variation in expression of gCIS target genes between tumor samples). Secondly, we performed Western analysis for two genes where predicted gain-of-function alleles had been created through SB-induced N-terminal truncation of the protein (*Notch1* and *Plakoglobin*). A novel smaller protein fragment of the predicted size was seen in both cases (Supplementary Figure 4).**

Comment 4

Most of the gCISs identified appear to target genes randomly, presumably resulting in “single allele disruption”. Because four of these genes are haploinsufficient tumor suppressors, the authors conclude that hemizygous loss of most of their genes results in “just enough and just right” oncogenic signaling pathway activation”. However, the authors present no expression validation for these statements. Again, (i) if hemizygous loss of genes has a functional impact, then gene expression differences and (ii) consequently differential activation of the respective oncogenic pathways should be observed and described in the manuscript. (iii) Moreover, in human cancer, loss of heterozygosity (through genetic or epigenetic mechanisms) is a common phenomenon, which especially affects tumor suppressors, such as *NF1* and *PTEN*. The authors should investigate whether LoH also takes place in their mouse models. If so, then their conclusion, hemizygous loss is “just enough and just right” for tumorigenesis, would be incorrect. **This point is related to comment 3 above. Beyond Nanostring and Western analysis as described, we have performed ddPCR based analysis of copy number in our validation experiments to test for haploinsufficient tumor suppressor gene activity (Figure 8). In crosses to test for *Nf1*, *Trps1* and/or *Fbxw7* hTSG activity we saw striking cooperation between *Nf1* hemizygous loss and *Trps1* hemizygous loss without evidence for copy number loss at either gene. In contrast, in assays to test *Nf1* and *Fbxw7* for cooperating hTSG activity, we saw statistically significant copy number loss at both loci - indicative of recessive tumor suppressor activity in this context. Thus, we have shown context dependent hTSG activity.**

Comment 5

In Figs. 5 and 6, the authors confirm loss and/or activation of several of the genes identified in their SB screens to be involved in tumorigenesis. However, the manuscript completely lacks any proof of functionality of these different tumor models. The authors should at least genotype the emerging tumors to demonstrate switching of the conditional alleles. Moreover, it appears important to again show hemizygosity on a gene expression level (compare to comments 3, 4) as much as protein expression of

induced genes. The authors should demonstrate cytoplasmic/nuclear localization of active CTNNB1dEx5, reduced expression of Nf1, Pten, Fbxw7, as well as expression of truncated Trps1dEx4. **We have now provide copy number analysis on tumors as described in Figure 8 and respectfully suggest that this is not feasible for other cohorts. Also, each mouse line used for experiments presented in Figure 6 are well characterized and have been used by many labs to show Cre-dependent gain-of-function or loss-of-function effects, as appropriate.**

Comment 6

One of the important conclusions of this manuscript is the following: If gCISs are present at clonal levels, then the gene alterations are likely tumor-initiating. In contrast, gCISs present at sub-clonal levels are representative of tumor progression events. I agree with clonal equals tumor initiation, while I disagree with sub-clonal equals tumor progression. It is as (if not far more) likely that sub-clonal detection of gCISs is a caveat of bulk-sequencing of heterogeneous tumors, which emerged from multiple cells of origin driven by different gCISs. Although the authors convincingly identify (clonal) gCISs that co-occur in certain tumor subtypes and/or driver gene contexts, sub-clonality versus clonality likely just represents weaker versus stronger driver capacities in these respective contexts. Weaker (therefore slower) drivers are co-emerging with other driving events, thus appearing sub-clonal, while strong drivers outcompete weaker drivers and appear clonal. **This is a very good point and we made a mistake by presenting the data in this way. We have now changed the text to correct this. On page 12, the new manuscript states “Functional genomic screens with SB, unlike screens using viruses like MMTV, involve mobilization of tens to hundreds of insertional mutagens (transposons) within each cell. As a result, this system has the potential to uncover or identify combinations of hemizygous loss-of-function mutations that cooperate in transformation. To test for this, we identified pairs of gCIS that showed higher-than-expected frequencies of co-occurrence. Such combinations conceivably represent cooperating genetic events within the same tumor cell, or events that cooperate with the driver in distinct cells (subclones) within the tumor.”** The efforts of the authors in Figs. 5 and 6 to confirm the tumor driving relevance of several of the gCIS hits via conditional inactivation of the gene (e.g. Nf1, Pten) are appreciated. However, unambiguous proof for relevance in tumor progression versus tumor initiation would require sequential induction of genes of interest (e.g. Pten loss) at a later timepoint (e.g. in established tumors) than the primary tumor driver (e.g. Pik3ca). Taken together, the authors should either rephrase their conclusions or demonstrate experimentally sub-clonal tumor progression versus multi-clonal tumors, the latter of which is challenging and might reach beyond the scope of this study. **Yes, we agree – see new text on page 12 discussed above.**

Comment 7

Data related to the human relevance of found screen hits should be presented in a figure instead of as Supp. Table 5. The authors should present statistical evidence whether candidate genes identified in their SB screens are indeed enriched on chromosome arms frequently hemizygously lost. **We have obtained the published human data from Dr. Beroukim and tested for this. There was no statistical enrichment for (or against) the inclusion of gCIS with regions of chromosome loss.** The authors could also elaborate on this further, for instance by looking at expression of their screen hits in RNA-seq datasets of human mammary tumor cohorts. **We respectfully suggest that such analysis, while interesting, is beyond the scope of our current study. Also, on the basis of our data, which shows selection for driver-specific gCIS in most cases, we would predict that reduced expression of any particular gene (from our gCIS list) would only occur in a situation where the same driver pathway is activated.** It would not be trivial to establish this for each gCIS under study. Along with this, the paragraphs describing comparison of screen hits to other transposon screens and to human breast oncogene and tumor suppressor genes should be either deleted from the manuscript (not very informative as table descriptions) or be presented as figure panels. **Unfortunately, the manuscript is 10 figures in length and therefore we have not been able to make additional figures or tables. We have now sharpened the text to highlight the novelty of our screen. We have made concise comparisons to other screens and to human data. This comparison is necessary to help understand why there are so many different gCIS uncovered by our study, and why there is a relatively small overlap between these genes and the 99 breast cancer gene list, which is composed mostly of dominant oncogenes and recessive tumor suppressors (Serena Nik-Zainal, Nature. 2016.**

534(7605):47-54). Indeed, the list of human breast cancer genes was derived on the basis of frequent SNV, focal and high copy number gains as well as focal homozygous gene loss. hTSG, buried within chromosome arm-length losses, are almost impossible to identify, except through functional genomic screens like ours. Recognizing that we didn't do a good job in explaining this, we have substantially rewritten this section. We have now included a small table (Table 1) to highlight gCIS whose orthologues map to chromosome arms that show hemizygous loss in approximately 50% (or more) of human breast cancers.

Comment 8

The different mouse strain abbreviations used are rather confusing. It would help the readability of the manuscript/figures, if gene names and gene variants were properly written out. Figure 2c and 4b have very limited value. It is unclear what all the dots and different colors mean. All the p-value tables are difficult to read, and should rather be presented as connecting lines on the plots itself. All the pie charts are difficult to read, especially in Figures 5-7. Shouldn't be referred to these by separate panel letters (a, b, c...)? In general, the Figures should be designed in a more appealing way. **We have taken this comment seriously and redone every figure. For example, all pie charts have been replaced by graphs. See comments in response to reviewer 2 below.**

Reviewer #2 (Remarks to the Author): Expert in mouse genetics

In their study Schachter et al. performed Sleeping Beauty insertional mutagenesis screens in eight different engineered (breast cancer predisposing) genetic backgrounds. This strenuous effort produced more than 700 tumours and a list of over 1000 common insertion sites. Although several Sleeping Beauty screens have been performed earlier in breast cancer, this is the first study to perform such a large-scale systematic comparison in a large number of genetic backgrounds using the same Cre line. This approach allows direct in vivo interrogation of a series of questions, including genetic interactions, the genetic basis of phenotypic diversification, oncogene dosage effects and others. Generally, the study is well done, involving scientists with long-standing expertise in the design, execution and analysis of transposon screens. The data demonstrate the power of the screening technologies to uncover biological principles and will certainly be a very valuable source of knowledge in the field.

However, I have major concerns about the presentation, the interpretation and discussion of the data, which is in large parts confusing and requires substantial revision. In almost all sections of the manuscript, it is impossible to understand which of the conclusions/discoveries are new? **We have tried to clarify this and to reduce the text in areas where some of the findings are consistent with pre-existing literature. Indeed, the manuscript has been substantially rewritten, with an attempt to clarify. Having said this, we quote pre-existing literature in order to credit and highlight the work of others as appropriate (see below).** In other words, the authors fail to convince the reader that the screens can uncover new biology, beyond what has been known before. As detailed below, I would suggest to clearly point out data that merely support known biology and to explain/present such data in a more condensed manner. Instead, the focus should be much more on the new discoveries, their interpretation, functional validation and discussion.

Specific comments:

A. Abstract

1. The abstract is a very poor summary of the scope, the results and conclusions of the study. In my eyes, it does not describe adequately the unique scale (700 cancers) of the study or the large number of biological questions that could in principle be interrogated (e.g. histological subtypes are missing). Instead, haploinsufficiency of Pten, Nf1, Fbxw7 and Trps1 is presented as a main discovery. For all four genes haploinsufficiency has been extensively reported before.

2. Main novel discoveries should be the guiding theme of the abstract

3. The “just enough and just right” concept of oncogenic signaling is put forward, without showing functional data. Although I support the interpretation, I feel that a more cautious wording would be more adequate to avoid overstatements (e.g. the data support the concept of “just enough and just right...”).

We have rewritten the abstract in an attempt to clarify and highlight our major conclusion, noting the scope of our study, the importance of histology and the central conclusion about selecting and sculpting, which is now better supported through the addition of new data (Figure 10).

New Abstract: The most common events in breast cancer (BC) involve loss or gain of chromosome arms. Transposon screens can be an effective way to identify single allele insertion mutations that contribute to cancer through copy number alterations (CNA). Here we describe identification of ~1000 gene-centric common insertion sites (gCIS) from transposon-based screens in 8 mouse models of BC from over 800 mouse tumors. Some gCIS were oncogene/tumor suppressor driver-specific, others driver non-specific, and still others associated with tumor histology. Processes altered by driver-specific mutations included well-described pathways controlling programmed cell death, chromatin organization, as well as RTK/MAPK and/or GTPase signal transduction and differentiation. Driver non-specific gCIS targeted autophagy, regulation of Ca⁺ ion concentration and chromosome localization. Most gCIS showed single allele disruption and many mapped to regions of the genome exhibiting high-frequency hemizygous loss in human BC. Two gCIS, when tested, *Nf1* and *Trps1*, showed conditional haploinsufficient tumor suppressor function. Finally, many gCIS act on the same pathway responsible for tumor initiation, thereby selecting and sculpting “just enough and just right” signaling. The large number of genes identified herein far exceeds the number previously implicated as targets of recurrent focal mutation, thereby highlighting the importance of low level CNA and the vast number of genes that contribute to cellular transformation when subject to hemizygous loss.

Previous Abstract: The most common events in breast cancer involve hemizygous loss or low-level copy number gain of chromosome arms. Here we describe identification of ~1000 gene-centric common insertion sites (gCIS) in transposon-based screens from 8 mouse models of BC. Most of these showed single allele disruption. When tested, four were conditional haploinsufficient tumors suppressors (hTSG)(*Nf1*, *Trps1*, *Fbxw7* and *Pten*). These and many other identified gCIS function on the same pathway responsible for tumor initiation. Thus, many genes that show hemizygous loss in breast cancer form conditional cooperative networks that select and sculpt “just enough and just right” oncogenic signaling pathway activation.

B. Results section 1 (page 4):

1. Overview. The study uses with a series of genetic backgrounds and crosses. It would be helpful to have an overview figure/table containing the different genotypes/crosses, number of mice, number of tumors and related histologies, number of CISs identified. **We have now included a figure to explain genotypes for each cohort (Supplementary Figure 1).**

2. In addition, because this manuscript represents a large resource, it would be useful to provide a supplementary table with detailed information for each individual mouse: (i) genotype, (ii) tumour number and histologies, number of insertions. **We have reworked every Supplementary table and figure in order to clarify the data. We have added two Supplementary Tables to provide the data requested by reviewer 2 (Supplementary Table 2 and Supplementary Table 4). Also, the primary data has been submitted for public access (GEO: GSE143503). While the oncoprint files (Supplementary Figures 3 and 7) do not list the specific mouse and tumor ID, they do show each tumor as a column whereby the number of insertions as well as the specific gCIS targeted in each tumor is shown. The data for each column in Oncoprint files can be found within Supplementary Table 2 and Supplementary Table 4.**

3. Transposon lines used and analyses. The authors state that most of the screens were performed using both the T2Onc3a und T2Onc3b transposon mouse lines (differing in the transposon donor chromosome). I could not find further information on this point. Were donor loci/chromosomes excluded from the analysis because of local hopping? How did the transposon line influence the data analyses. **Onc3a and Onc3b cohorts were kept separate, and the two different concatemers effected tumor latency, penetrance and even tumor histology differently (see Supplementary Figures 2 and 6). We do not understand why this is so. Donor chromosomes were removed from the analysis. We should have stated this explicitly. We have now included a statement in the Methods section (SB read preprocessing, alignment and analysis subsection) that makes this point clear “All insertion sites mapping to mouse chromosome 9 and 12 were filtered out so as not to misidentify local hopping events as gCIS”.**

4. Data analyses. Clonal vs. subclonal insertions: Although cutoffs for clonal and subclonal insertions are mentioned, the details remain unclear (but are important for an in depth understanding). How were the cutoffs defined? **In the Methods section on SB read preprocessing, alignment and analysis, the following statement is made “A dynamic filter was used to categorize insertions as clonal or subclonal. For each library, three thresholds were calculated using the insertion data: (i) > 95% percentile of reads under the negative binomial distribution, (ii) 1% of the most abundant insertion sites, (iii) 0.1% of the total reads.”** How many clonal/subclonal insertions did the authors find per tumor? How does the distribution look like? Were differences in clonality detected? **In each Oncoprint file there are three pieces of information provided to address these questions (Figures 1, 3, Supplementary Figure 3 and 7). In the legend for Figure 1, this is now described in detail – “Note, the percentage of tumors with each gene targeted by SB is shown on the y-axis to the left for each bar graph, the number of tumors with clonal vs. subclonal SB targeting by SB is shown on the y-axis to the right, whereas the number of identified gCIS in each tumor is shown on the x-axis above each bar graph.**

C. Results sections 2 (page 5) and 3 (page 7):

1. Supplementary figure 2 contains important information and should be presented in the main manuscript, by adding the plots to Figure 1. **While it is not possible to include complete gCIS data within the primary figures, we agree with the spirit of this comment and have now redone Figure 1 to show the top hits for each driver specific screen.**

2. These sections show extensive context-specificity of the CISs identified in the study: in different crosses/predisposing backgrounds CIS lists (genes and related pathways) differ substantially. The data impressively demonstrate the power of insertional mutagenesis to uncover biological principles. The authors list a series of examples of how the screen pinpoints for example cooperativity between genes and pathways. However, it remains unclear what is new and what has been known before (particularly for readers who are not experts in breast cancer biology). It is essential to rewrite these two chapters (but also many of the subsequent ones) in order to:

(i) discuss data related to known biology and associations in a more condensed form, and
(ii) point out new discoveries/biology, for which the authors should provide in depth functional experiments/data (at least for one example). In addition, to help readers navigate through the the large and complex data sets, it would help to end each section of the manuscript with one or two sentences summarizing the main findings and putting them into context. **We agree with this point and have rewritten these sections in every effort to address this incredible challenge. The very study design, a large scale transposon-based cancer gene discovery screen in the mouse mammary gland, necessarily yields a mixture of data that is new and insightful with data that confirms what we know, either from previous functional genomic screens in the mouse mammary gland or from human breast cancer genomics. Some level of high level discussion is required to put the data in context. We have completely rewritten the manuscript, so much so that it is not possible to submit a track changes file that can be understood. We have shortened some of the presentation where our results are more confirmatory (see discussion of Wnt and Notch below). As noted, we have made every effort to quote pre-existing literature in order to credit and highlight the work of others as appropriate.**

3. Figure 2c and 4b (pathway analyses). The figures display important findings, but are very hard to comprehend because (i) dots are too small and (ii) it seems that colours and labels have been mixed up. For example, in figure 2c Stat3c should be pink (see legend) but the dots in the pathways which are found in the Stat3c cohort (Pi3k or Wnt, see main text) are not pink. Likewise, Fgfr/Toll-like and IL signaling are exclusive for Stat3c according to the text but are colored dark blue, which corresponds to PIK3CA HR according to the legend. **These figures have been completely redone to sharpen them, to make them correspond in colour to other figures, and to enhance comprehension of our findings.**

D. Results sections 4 (page 8) and 5 (page 9).

1. These two sections aim at establishing a link between genetics and histopathology. The data are important. I also like the presentation of data in figure 3a. However, the interpretation and discussion is confusing. It is again impossible to understand whether the authors were able to use the data in order to uncover new biology, beyond what has been known before. After reading the chapters multiple times, one remains confused. For example, the authors focus on Notch, PI3K and Wnt signaling for downstream

analyses and validation. A Pubmed entry for breast cancer displays 800, 1900 and 3700 publications on Notch, Wnt or PI3K signaling, respectively. The link between papillary tumours and Notch1, for example, has been known before and it seems that the link between Wnt and the more basal subtypes is also common knowledge. We have shortened the discussion on Notch and Wnt and now highlight a new model whereby Wnt/Notch signaling antagonistically define tumor cell identity along a skin-basal-luminal cell fate determination axis. The experiment on Notch1 as tumor suppressor, the most novel finding here is described in this context, a reveal separation between oncogenic roles and cell-fate roles of these signaling pathways. The shorted discussion on gain-of-function Notch1^{ICD} and β -catenin is necessary to validate our data but also, importantly, to set up the model for cell fate determination that is tested with Notch1 deletion.

2. The authors cross Notch1-ICD and PI3K mutant mice. Based on this experiment they suggest that there is collaboration between Notch1-ICD and PI3K alleles in induction of papillary histology. However, the penetrance of papillary tumours is already 100% in the Notch1-ICD model. This is true in the sense that Notch1^{ICD} is a major driver of papillary cell fate when tumors form. However, Notch1^{ICD} by itself is very weak. The few tumors that form take a very long time to develop, almost certainly because they need to activate PI3K signaling for tumor cells to survive and grow. Interestingly, Notch1^{ICD} forms transient pregnancy dependent adenomas that disappear with weaning (PMID: 15277242). This is almost certainly due to the high level of Igf2 signaling in alveolar cells during pregnancy (PMID: 12479812). The two papers quoted above are important but quite old and do not address the specific importance of PI3K signaling.

3. Page 10: "Notch1 can function as an oncogene in cooperation with *Pik3ca*E545K and *Pik3ca*H1047R, but only as an allele-specific tumor suppressor in cooperation with the latter". This finding is interesting. What is the interpretation? At a detailed level, we do not understand the mechanism responsible for allele specific cooperation between *Pik3ca* hotspot mutants and *Notch1* deletion. However, in the revised manuscript we discussed the issue in context of the literature and results from our screen "While *Pik3ca*^{H1047R} cooperated with *Notch1*^{loxP/loxP} to decrease tumor latency, *Pik3ca*^{E545K} did not (Fig. 6c). Thus, in mammary epithelium, *Notch1* can function as an oncogene in cooperation with *Pik3ca*^{E545K} and *Pik3ca*^{H1047R}, but as an allele-specific tumor suppressor gene in cooperation with *Pik3ca*^{H1047R}. The reason for allele specificity with respect to *Notch1* tumor suppressor gene function in this context is unclear. However, the lack of cooperation between *Notch1* gene loss and expression of *Pik3ca*^{E545K} does not appear to affect the ability of *Notch1* deletion to skew tumors towards a squamous fate, revealing a separation between the ability of alterations in the Wnt/Notch signaling axis to promote transformation from their ability to effect tumor histology. In addition, the striking difference between *Pik3ca*^{E545K} and *Pik3ca*^{H1047R} in this assay is consistent with the very different list of gCIS identified in our SB screen for each allele (of 32 *Pik3ca*^{E545K}-derived and 62 *Pik3ca*^{H1047R}-derived gCIS, only 6 were identified in both screens (Supplementary Fig. 3 and Supplementary Table 1: *Met*, *Jup*, *Axin1*, *Myst3/Kat6a*, *Rab1* and *Tm9sf3*)). Thus, our results are consistent with the notion that elevated *Notch1* signaling drives tumor cell differentiation towards the luminal fate, whereas decreased *Notch1* signaling or elevated Wnt signaling promotes tumor differentiation towards a more basal, skin or pluripotent fate.

E. Results sections 6 (page 11) on cancer gene cooperation.

1. Cooperative haploinsufficient tumor suppressors: "We identified pairs of gCIS that showed higher-than-expected frequencies of co-occurrence" (p 11 line 9). Which other pairs besides *Nf1*-*Trps1* and *Nf1*-*Fbxw7* were detected? How many pairs were found to be statistically significant? There was a large number that showed statistical co-occurrence (over 2500 pairs in the large cohort). Based on ddPCR results shown in Figure 8, it's clear that some of these pairs will represent co-selected mutually-conditional haploinsufficient tumor suppressor gene pairs like *Nf1* and *Trps1*, whereas others could represent genes selected in different subclones (as suggested by both reviewers). Were these pairs found in combined analyses (across cohorts) or are there background specific pairs? Some pair were co-selected in individual driver-specific cohorts, whereas others were only seen in the large cohort (which had more statistic power to identify co-selection). *Nf1* and *Trps1* were statistically co-selected in the large cohort but not in the smaller *Pik3ca*^{H1047R} cohort.

2. Figure 6, cooperation between monoallelic loss of *Nf1*-*Trps1* and *Nf1*-*Fbxw7*. Does LOH occur in tumors? In addition, the authors might consider speculating about the biology of this collaboration in their discussion. We have now performed ddPCR analysis and understand *Nf1*^{+/-}:*Trps1*^{+/-} cooperation vs

Nf1+Fbxw7 cooperation very differently. We have decided not to speculate on the basis for this cooperation because we don't have a really compelling model to offer.

F. Results section 7 (page 12):

1. Minor comment: The authors might consider mentioning that potential confounders of analyses linking clonal/subclonal insertions to early/late tumorigenesis are not considered here, e.g. when an early driver insertion is not needed during later progression, transposons will jump out of the locus in many cells. Clonal sweep is another example. **We have chosen not to discuss the issue of oncogene-addiction or related concepts as we don't have any data to speak to it. This is an important issue and may well be relevant to our system, we just can't tell from our analysis (there doesn't seem to be a great deal of evidence for remobilization of SB once it has moved from concatemer to individual TA target sites in the genome).**

G. Results section 8 (page 13).

1. Minor: please include the term "human" to avoid confusion: Identification of candidate drivers behind human chromosome loss. **We have now done this.**

A table, rather than long lists of genes in the main text might improve readability. More generally, the authors state that "many [identified hTSGs] map to chromosome arms showing hemizygous loss [in breast cancer]" (p13 last line). Although some examples are given, it remains unclear what "many" means. Is there a statistically significant enrichment in these regions compared to others? **We have now included a small table (Table 1) to highlight gCIS whose orthologues map to chromosome arms that show hemizygous loss in approximately 50% (or more) of human breast cancers. However, as noted above, there was no statistical enrichment for (or against) the inclusion of gCIS with regions of chromosome loss.**

H. Results section 9 (page 14).

1. To me this is the most fascinating part of the study: selection for insertions in genes affecting the pathway of the initiating engineered oncoprotein. Unfortunately, the authors do not show any functional data. However, they present in the previous section that Pten loss cooperates with PI3K. I would suggest to move those data into this section to strengthen this important aspect of the study. **We now present a more complete analysis of this point based on analysis of an allelic series of *Pik3ca* hotspot mutations in syngeneic mice (Figure 10)**

2. Minor: The authors discuss the importance of oncogenic dosage variation in cancer only in a model in which an oncogene/pathway is co-regulated/modulated by other genes/pathways. They might consider mentioning another model, the dosage variation of the oncogene itself (e.g. PMID: 29364867, for example through copy number gain), which is a widespread phenomenon. It strengthens the point of the whole section as it provides compelling evidence for the validity of the "just enough and just right" concept that the authors put forward. **This is a great suggestion. Indeed, we have added a whole set of experiments based on testing this concept (Figure 10). We now cite the K-Ras paper noted by Reviewer 2 in the discussion section.**

I. Further minor remarks:

In several figures, the authors might consider improving data presentation to increase accessibility (labeling, colors, style, quality, readability). In addition, there are a number of mistakes that need to be corrected. **We have remade every figure.** For example:

1. Labeling of cohorts: using a consistent labeling of the cohorts would lead to an improved understanding (the Notch1 cohort is labeled as Notch1 ICD in Figure 1/2/4, as RN1 in Figure 3 and as N1-ICD in Figure 5. Comparable for p53 R270H which is sometimes only labeled as R270H). **We have improved the labeling and clearly defined any abbreviations which were required to use in order to save space in some places.**

2. Using the same colors for the cohorts in Figure 2 and 3 would make it more comprehensive. **We have now redone the figures to make the colours match.**

3. Figure 4a: For easier comparison of the histological subtypes between cohorts I would suggest a bar graph representation with only one legend. **We have now changed pie charts to bar graphs.**

4. Figure 4b: More colors in the dots than in legend. There are multiple yellow dots in the Figure but yellow is not described in the legend. **This has been fixed.**

5. Legends of pathway analysis figures are very blurred and some dots too small to see different colors. **These have been redone and are now sharp.**
6. Figure 6: Top and middle panel are not labelled correctly (change Figure legend). Text and figure legends:
 1. Why are the two cohorts of Pik3ca summarized as only one in Figure 3 although tumor spectrum seems to be different according to Figure 4? **With our previous submission, pathway analysis was performed on Pik3ca^{H1047R} tumors but not Pik3ca^{E545K} tumors. Both have now been analyzed and the data presented as Pik3ca^{E545K + H1047R} for simplicity (detailed pathway analysis is provided in Supplementary Table 3).**
 2. Figure legend Figure 4: “Pathways altered by SB mutagenesis in GEMM-specific subscreens”: copy and paste mistake from Figure 2c. **This has been fixed in the new figure legend (for Figure 5).**
 3. Figure legend Figure 5: three identical combinations of Kaplan-Meier curves and pie charts are labelled as:
 - a) Kaplan-Meier survival analysis (top) and histotypes of mammary lesions (bottom)
 - b) Survival analysis (left) and pie charts (bottom)
 - c) Survival analysis (top) and pie charts (bottom) **This redundancy has been removed.**
4. Introduction, page 3: “Although small scale SB screens have been performed...” The cited screens were mostly adequately sized for the questions that they addressed. **The description of these screens has been changed and the word small removed.**

REVIEWERS' COMMENTS

Reviewer #2 (Remarks to the Author):

All my major concerns have been addressed. I recommend publication.